# A confirmation bias in perceptual decision-making due to hierarchical approximate inference

**Richard D. Lange**[1,2¤a]\*, **Ankani Chattoraj**[1], **Jeffrey M. Beck**[3], **Jacob L. Yates**[1¤b], **Ralf M. Haefner**[1,2]\*

**1** Brain and Cognitive Sciences, University of Rochester, Rochester, New York, United States of America, **2** Computer Science, University of Rochester, Rochester, New York, United States of America, **3** Department of Neurobiology, Duke University, Durham, North Carolina, United States of America

¤a Current address: Department of Neurobiology, University of Pennsylvania, Philadelphia, Pennsylvania, United States of America
¤b Current address: Department of Biology, University of Maryland, College Park, Maryland, United States of America

\* lange.richard.d@gmail.com (RDL); ralf.haefner@gmail.com (RMH)

**Data Availability Statement:** Location of data and code: https://osf.io/mxw5v/.

**Funding:** This work was supported by National Eye Institute/NIH awards R01 EY028811 (RMH) and

## Abstract

Making good decisions requires updating beliefs according to new evidence. This is a dynamical process that is prone to biases: in some cases, beliefs become entrenched and resistant to new evidence (leading to primacy effects), while in other cases, beliefs fade over time and rely primarily on later evidence (leading to recency effects). How and why either type of bias dominates in a given context is an important open question. Here, we study this question in classic perceptual decision-making tasks, where, puzzlingly, previous empirical studies differ in the kinds of biases they observe, ranging from primacy to recency, despite seemingly equivalent tasks. We present a new model, based on hierarchical approximate inference and derived from normative principles, that not only explains both primacy and recency effects in existing studies, but also predicts how the type of bias should depend on the statistics of stimuli in a given task. We verify this prediction in a novel visual discrimination task with human observers, finding that each observer's temporal bias changed as the result of changing the key stimulus statistics identified by our model. The key dynamic that leads to a primacy bias in our model is an overweighting of new sensory information that agrees with the observer's existing belief—a type of 'confirmation bias'. By fitting an extended drift-diffusion model to our data we rule out an alternative explanation for primacy effects due to bounded integration. Taken together, our results resolve a major discrepancy among existing perceptual decision-making studies, and suggest that a key source of bias in human decision-making is approximate hierarchical inference.

## Author summary

When humans and animals accumulate evidence over time, they are often biased. Identifying the mechanisms underlying these biases can lead to new insights into principles of

T32 EY007125 (RDL,JLY), as well as an National Science Foundation/NRT graduate training grant NSF-1449828 (RDL). The funders had no role in study design, data collection and analysis, decision to publish, or preparation of the manuscript.

**Competing interests:** The authors have declared that no competing interests exist.

neural computation. The confirmation bias, in which new evidence is given more weight when it agrees with existing beliefs, is a ubiquitous yet poorly understood example of such biases. Here we report that a confirmation bias arises even during perceptual decision-making, and propose an approximate hierarchical inference model as the underlying mechanism. Our model correctly predicts for what stimuli and tasks this bias will be strong, and when it will be weak, a critical prediction that we confirm using old and new data. A quantitative model comparison clearly favors our model over a key alternative: integration to bound. The key dynamic driving the confirmation bias in our model is an interaction between inferences on different timescales, a common scenario in decision-making more generally.

## Introduction

Human decisions are known to be systematically biased, from high-level planning and reasoning to low-level perceptual decisions [1, 2]. Decisions are especially difficult when they require synthesizing multiple pieces of noisy or ambiguous evidence for or against multiple alternatives [3–6]. Perceptual decision-making studies across multiple species and sensory modalities have exposed systematic biases that differ in ways that are not well understood. Here, we focus on temporal biases, which range from over-weighting early evidence (a primacy effect) to over-weighting late evidence (a recency effect) (Fig 1A) even in situations when an equal weighting of evidence would be optimal. Despite seemingly comparable tasks, existing studies are surprisingly heterogeneous in the biases they find: some report primacy effects [7–9], some find that information is weighted equally over time [10–12], and some find recency effects [13] without a clear pattern emerging from the data.

Existing models propose mechanisms for either primacy [7] or recency [14] effects alone, or are flexible enough to account for either type of bias [3, 4, 11, 13, 15–18], but none identifies or predicts factors that cause one bias or the other to appear in a given experimental context. All of these models are based on a variant of the classic drift-diffusion model [5]. For example, Kiani et al (2008) proposed that evidence integration stops when an internal bound is reached, even during fixed stimulus duration tasks. Averaged over many trials in which the bound is reached at different times, this leads to a primacy effect. Alternatively, a primacy effect is also expected if evidence integration is implemented by an attractor network [16, 17, 19], or mutual inhibition of competing accumulators [15, 18, 20]. However, neither of these mechanisms can account for recency effects. On the other hand, including a "forgetting" or "leak" term in the updating of the decision-variable leads to a recency effect [3, 4, 14, 15, 17, 18, 20]. The analysis by Glaze et al (2015) shows that a recency bias is optimal in a volatile environment, but such mechanisms cannot explain primacy effects [14]. Deneve (2012)'s normative analysis predicts that primacy and recency should depend on trial-by-trial changes in difficulty [21], while Prat-Ortega et al (2021) find that primacy and recency can change as a function of the variability of the input to a attractor-based decision-circuit [22]. However, neither account alone, or in combination, can explain the differences found across experiments. It is thus an open question whether the disparate biases observed empirically are due to differences in species, sensory modalities, training, experimental design, or individual observers.

Here, we propose a new model that not only accounts for the existing findings in the literature, but also predicts which key aspect of the stimulus determines the specific temporal bias shown by an observer. Our model extends classic ideal observer models to the hierarchical case by explicitly including the intermediate sensory representation. This reveals that task

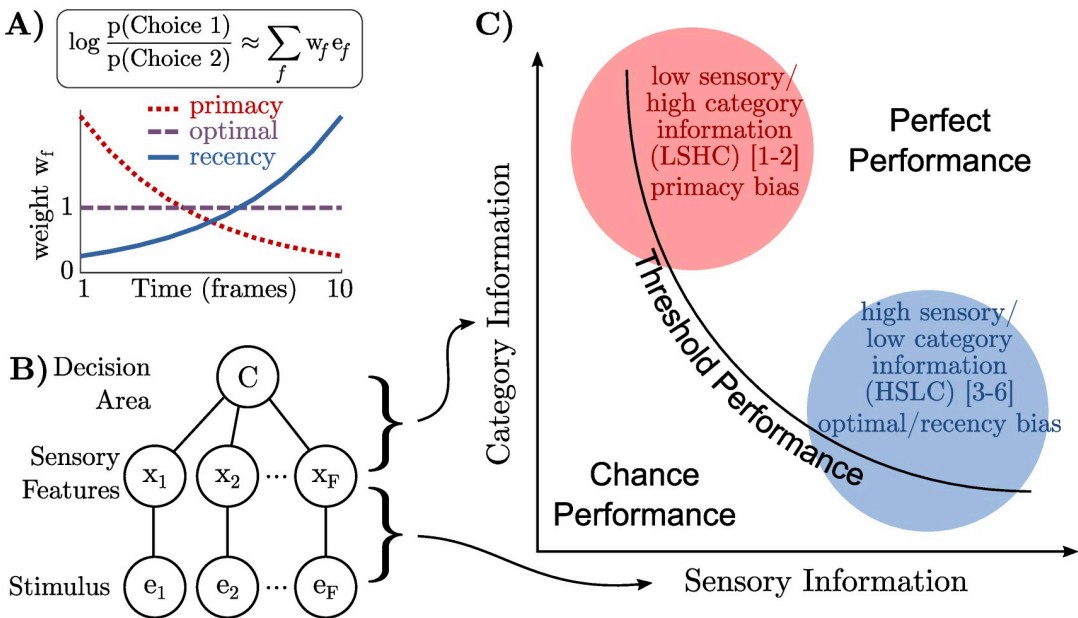

**Fig 1. Differences in "sensory information" and "category information" can explain differences in temporal biases reported by earlier studies. a)** A observer's "temporal weighting strategy" is an estimate of how their choice is based on a weighted sum of each frame of evidence $e_f$ (more precisely, a weighted sum of the log odds at each frame). Three commonly observed motifs are decreasing weights (primacy), constant weights (optimal), or increasing weights (recency). **b)** Information in the stimulus about the category may be decomposed into information in each frame about a sensory variable ("sensory information") and information about the category given the sensory variable ("category information"). **c)** Category information and sensory information may be manipulated independently, creating a two-dimensional space of possible tasks. Any level of task performance can be the result of different combinations of sensory and category information. A qualitative placement of previous work into this space separates those that find primacy effects in the upper-left (low sensory/high category information or LSHC regime) from those that find recency effects or optimal weights in the lower right (high sensory/low category information or HSLC regime). Numbered references are: [1] Kiani et al (2008), [2] Nienborg and Cumming (2009), [3] Brunton et al (2013), [4] Wyart et al (2012), [5] Raposo et al (2014), [6] Drugowitsch et al (2016). See S1 Text for justifications of placements.

difficulty is modulated by two distinct types of information: the information between the stimulus and sensory representation ("sensory information"), and the information between sensory representation and category ("category information") (Fig 1B). We show that approximate inference in such a model predicts characteristic temporal biases in a way that can explain prior empirical findings. Furthermore, our model makes a critical prediction: that the temporal bias of an individual observer should change from primacy to recency as the balance in the types of information is changed. We verify this critical prediction of our model using newly collected data from a novel pair of visual discrimination tasks. Finally, we perform a quantitative model comparison demonstrating that inference dynamics, not a finite integration bound, explain our observers' biases, consistent with our theory.

## Results

### "Sensory information" vs "Category information"

Normative models of decision-making in the brain are typically based on the idea of an *ideal observer*, who uses Bayes' rule to infer the most likely category on each trial given the stimulus. On each trial in a typical task, the stimulus consists of multiple "frames" (by "frames" we refer to independent pieces of evidence that are not necessarily visual). If the stimulus or evidence

in each frame, $e_f$, is independent, then information about the category $C$ can be combined by the well-known process of summing the log odds implied by each piece of evidence [4, 5, 23]:

$$\underbrace{\log \frac{p(C=+1|e_1,\ldots,e_F)}{p(C=-1|e_1,\ldots,e_F)}}_{\text{Log Posterior Odds, LPO}_F} = \underbrace{\log \frac{p(C=+1)}{p(C=-1)}}_{\text{Log Prior Odds}} + \sum_{f=1}^{F} \underbrace{\log \frac{p(e_f|C=+1)}{p(e_f|C=-1)}}_{\text{Log Likelihood Odds, LLO}_f}. \qquad (1)$$

The ideal observer updates their current belief about the correct category by adding the information provided by the current evidence: $\text{LPO}_f = \text{LPO}_{f-1} + \text{LLO}_f$.

In the brain, however, a decision-making area cannot base its decision on the externally presented stimulus, $e_f$, directly, but must rely on intermediate sensory features, which we call $x_f$ (Fig 1B). Accounting for the intervening sensory representation implies that $\text{LLO}_f$ cannot be computed directly, but only in stages. The information between the stimulus and category ($e_f$ to $C$) is therefore partitioned into two stages: the information between the stimulus and the sensory features ($e_f$ to $x_f$), and the information between sensory features and category ($x_f$ to $C$). We call these "sensory information" and "category information," respectively (Fig 1B). These two kinds of information define a two-dimensional space in which a given task is located as a single point (Fig 1C). For example, in a visual task, each $e_f$ would be the image on the screen while $x_f$ could be the instantaneous orientation or motion direction.

An evidence integration task may be challenging either because each frame is perceptually unclear (low "sensory information"), or because the relationship between sensory features and category is ambiguous in each frame (low "category information"). Consider the classic dot motion task [24] and the Poisson clicks task [11], which occupy opposite locations in the space. In the classic low-coherence dot motion task, observers view a cloud of moving dots, a small percentage of which move "coherently" in one direction. Here, sensory information is low since the percept of net motion is weak on each frame. Category information, on the other hand, is high, since knowing the true net motion on a single frame would be highly predictive of the correct choice (and of motion on subsequent frames). In the Poisson clicks task, on the other hand, observers hear a random sequence of clicks in each ear and must report the side with the higher rate. Here, sensory information is high since each click is well above sensory thresholds. Category information, however, is low, since knowing the side on which a single click was presented provides only little information about the correct choice for the trial as a whole (and the side of the other clicks). When frames are sequential, another way to think about category information is as "temporal coherence" of the stimulus: the more each frame of evidence is predictive of the correct choice, the more the frames must be predictive of each other, whether a frame consists of visual dots or of auditory clicks. Note that our distinction between sensory and category information is different from the well-studied distinction between internal and external noise; in general, both internal and external noise will reduce the amount of sensory and category information.

In general, sensory and category information depends on the nature of the sensory features **x** relative to $e$ and $C$, and those relationships depend on the sensory system under consideration. For instance, a high spatial frequency grating may contain high sensory information to a primate, but low sensory information to a species with lower acuity. Similarly, when "frames" are presented quickly, they may be temporally integrated, with the effect of both reducing sensory information and increasing category information.

Qualitatively placing prior studies in the space spanned by these two kinds of information results in two clusters: the studies that report primacy effects are located in the upper left quadrant (low-sensory/high-category or LSHC) and studies with flat weighting or recency effects are in the lower right quadrant (high-sensory/low-category or HSLC) (Fig 1C; see S1 Text for

justifications of placements). This provides initial empirical evidence that the trade-off between sensory information and category information may underlie differences in temporal weighting seen in previous studies. Unfortunately, since our placement of prior studies is only qualitative this observation only constitutes weak evidence in favor of this hypothesis. However, this hypothesis makes the strong prediction that a simple change in the stimulus statistics corresponding to sensory and category information, while holding everything else constant, should change the temporal weighting found in these previous studies (predictions provided in Table A in S1 Text). Below we will present new data from an experiment in which we did exactly that and found that biases indeed shifted from primacy to optimal/recency as predicted.

## Approximate hierarchical inference explains transition from primacy to recency

If stimuli were processed by the brain in a purely feedforward fashion, then a decision-making area could simply integrate the evidence in sensory features ($x_f$) directly. This is consistent with some theories of inference in the brain in which sensory areas represent a likelihood function over stimuli [25–28]. However, activity in sensory areas does not rigidly track the stimulus, but is known to be influenced by past stimuli [29, 30], as well as by feedback from the rest of the brain [31, 32]. In fact, the intermediate sensory representation is itself often assumed to be the result of an inference process over latent variables in an internal model of the world [33–35]. This process is naturally formalized as *hierarchical inference* (Fig 2A) in which feedforward connections communicate the likelihood and feedback communicates the prior or other contextual expectations, and sensory areas combine these to represent a posterior distribution [27, 36–39].

We hypothesize that feedback of "decision-related" information to sensory areas [40, 41] implements a prior that reflects current beliefs about the stimulus category [39, 42, 43]. While such a prior is optimal from the perspective of estimating the sensory features, $x_f$, this complicates evidence accumulation (Methods). When $x_f$ is influenced by prior beliefs about the stimulus category, the calculation of the "update" (log likelihood odds or LLO$_f$) cannot simply replace p($e_f|C$) by p($x_f|C$); instead, the decision-making area would need to account for or "divide out" the influence of the top-down prior on the sensory representation to avoid a double-counting of the prior (Fig 2B and 2C). For an ideal observer performing exact inference, this process would not entail any suboptimalities or biases. However, inference in the brain is necessarily approximate, with the *potential* to induce a bias.

*Under*-correcting for this prior would lead to earlier frames entering into the update multiple times, giving them a larger weight in the final decision. Over multiple frames, the effect is a positive feedback loop between estimates of sensory features $x_f$ and the belief in *C*. This mechanism constitutes a "perceptual confirmation bias," since belief updates are biased towards confirmatory evidence [2, 44], and leads to primacy effects. *Over*-correcting for the prior, on the other hand, would lead to information from earlier frames decaying away, giving earlier frames less influence on the final decision and manifesting as recency effects. In either case, the strength of any bias is directly related to the strength of the prior.

Importantly, the strength of the prior only depends on the amount of category information, and *not* the amount of sensory information (unlike task performance which depends on both). For instance, in a task with high category information such as the classic dot motion task [24], high certainty in the trial category, based on stimulus frames seen so far within a trial, directly translates into high certainty about the net motion direction in subsequent frames of that trial (Fig 2B). In a low category information task such as the Poisson clicks task [11], on the other

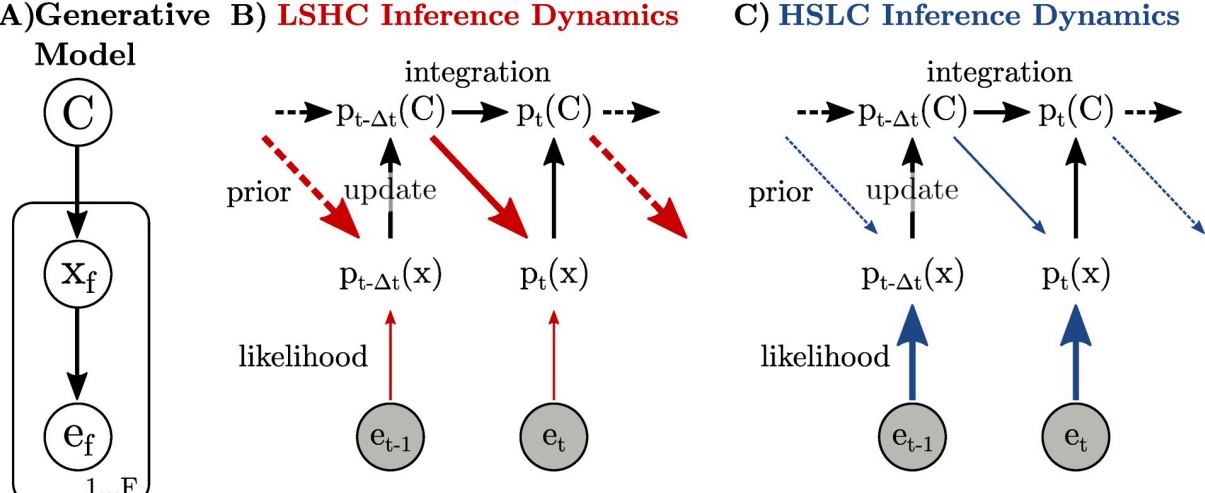

**Fig 2. Information flow during hierarchical inference where categorical beliefs are fed back as a prior on sensory features. a)** Generative model that we assume the brain has learned for a discrimination task, which specifies how sensory observations $e_f$ depend on the category for the trial, $C$, in two stages: each sensory observation $e_f$ is assumed to be a noisy realization of underlying sensory features, $x_f$, and each frame of sensory features is itself assumed to be selected according to the trial's category. **b-c)** Integrating evidence about $C$ requires updating the current belief about $C$ with new information derived from the sensory representation (left-right "integration" and bottom-up "update" arrows). The posterior distribution over $x$ combines top-down expectations (diagonal "prior" arrows) with new evidence from the stimulus, $e_f$ (bottom-up "likelihood" arrows). Width of arrows indicates average amount of information communicated; red and blue arrows indicate changes in information flow between conditions. Note that when inference is *exact*, the prior is subtracted from the information in the update during the integration to prevent double-counting early evidence. While the generative model in (a) operates with discrete frames, $f$, inference in the brain happens in continuous time, $t$. **b)** LSHC: Low sensory information means little information in the likelihood about sensory features $x_f$. High category information means that most of this information is also informative about $C$. It also means high information in the prior that is fed back to the sensory representation. **c)** HSLC: High sensory information means high information in the likelihood about sensory features $x_f$. Low category information means that this information is only weakly predictive of $C$. It also means little information in the prior that is being fed back to the sensory representation.

hand, certainty in the trial category is only weakly predictive whether any one click is on the left or on the right (Fig 2C). In the motion dots task, the relevant sensory feature, *x*, is the *net* motion of all dots, not the motion of any one dot. The net motion in later frames is highly predictable from the net motion in earlier frames (i.e. high category-information) even if the motion of any one dot is not, and it is known that the sensory neurons involved in the task represent the net motion by averaging over the motion of many neurons within their receptive field [5].

We implemented two canonical models, corresponding to each of the two major classes of approximate inference schemes known from statistics and machine learning: sampling-based and variational inference [45, 46], and both of which have previously been proposed models for inference in the brain [27, 36, 37, 47]. In both cases, we assumed that sensory areas of the brain approximate the posterior, incorporating both the current sensory input *and* expectations based on past frames. Interestingly, both sampling-based and variational inference models behaved similarly in terms of performance and biases, and so here only show the results from the sampling-based model, and provide the corresponding variational results in the SI. The performance of our approximate models (Fig 3B) largely matched that of an exact inference model (Fig 3A), with accuracy somewhat reduced for high category information. We computed the temporal biases of the approximate inference models for each combination of sensory information and category information, and found that both models showed a primacy effect whose magnitude decreased with the amount of category information (Fig 3B, 3C and

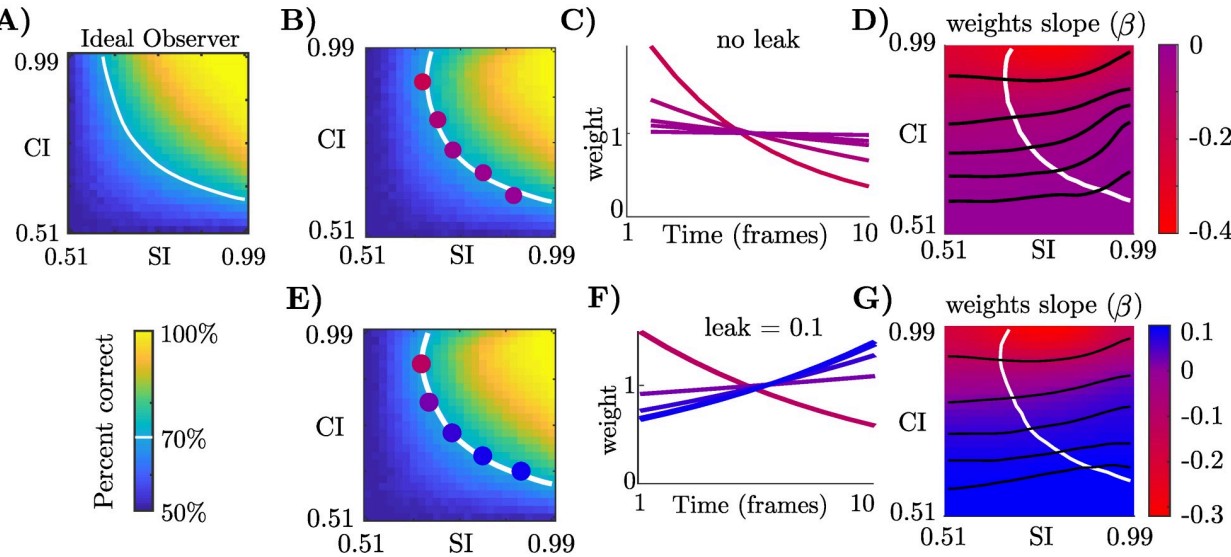

**Fig 3. Changes in bias predicted by approximate hierarchical inference models. a)** Performance of an ideal observer reporting *C* given ten frames of evidence. White line shows threshold performance, defined as 70% correct. The ideal observer's temporal weights are always flat (not shown). **b)** Performance of our sampling-based approximate inference model with no leak (Methods). Colored dots correspond to lines in the next panel. **c)** Temporal weights in the model transition from flat to a strong primacy effect, all at threshold performance, as the stimulus transitions from the HSLC to the LSHC conditions. **d)** Visualization of how temporal biases change across the entire task space. Red corresponds to primacy, and blue to recency. White contour as in (b). Black lines are iso-contours for slopes corresponding to highlighted points in (b). **e-g)** Same as (b-d) but with leaky integration, which lessens primacy effects and produces recency effects when category information is low.

3D). Both hierarchical inference models *under*-corrected for the influence of prior expectations on the sensory representation. Over the course of a trial, this lead to a positive feedback loop between the evidence-integration part of the model and the sensory representation, with the strength of this loop being strong in the LSHC and weak in the HSLC condition (Fig 2B and 2C). Importantly, this bias is a direct consequence of the *approximate* nature of the representation of the posterior distribution; for instance, in the sampling model, the bias disappears as the number of samples gets large (Methods).

Results for the variational and sampling-based inference models are qualitatively the same (Fig D in S1 Text), as are results from simulating a larger neurally-inspired sampling model (Fig H in S1 Text) [42]. This indicates that the observed pattern of biases is not tied to a particular representation scheme—sampling or parametric—but to the *approximate* and *hierarchical* nature of inference.

Previous studies further suggest that evidence integration in the brain may be "leaky" or "forgetful," which can be motivated either mechanistically [3, 4, 13, 17], or as an adaptation to non-stationary environments in normative models [14]. Including leaky integration, our final approximate inference models contain two competing mechanisms: first, they exhibit a confirmation bias as a consequence of approximate hierarchical inference, which is strongest when category information is high, leading to a primacy effect. Second, they contain leaky integration dynamics, which dampens the primacy effect and results in recency effects when category information is low and confirmation bias dynamics are weak (Fig 3E, 3F and 3G). While the exact magnitude of the leak is a free parameter in our model, to be constrained by data, the *change* in bias with changes in category information is a strong prediction, i.e. changing from strong primacy to no bias, or changing from weak primacy to recency, depending on the magnitude of the leak.

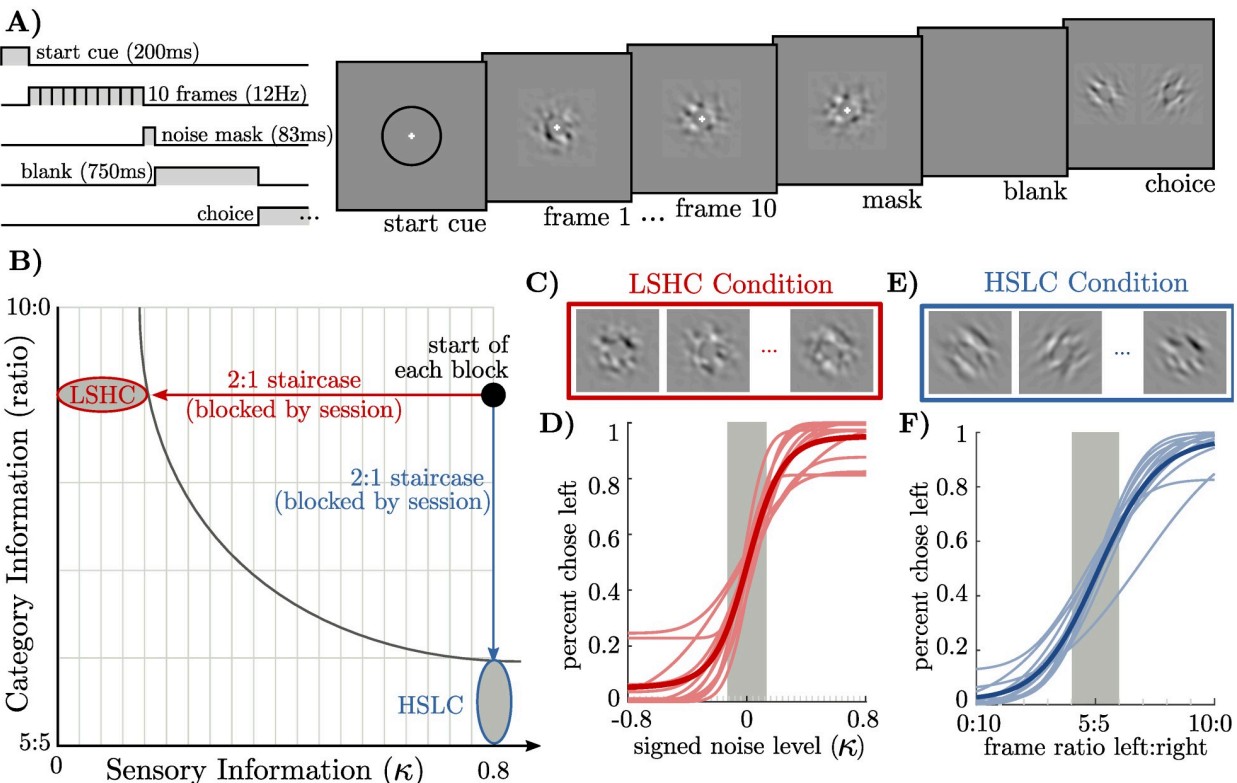

**Fig 4. Two task conditions that reduce either sensory or category information to threshold level using a staircase. a)** Each trial consisted of a 200ms start cue, followed by 10 stimulus frames presented for 83ms each, followed by a single mask frame of zero-coherence noise. After a 750ms delay, left or right targets appeared and participants pressed a button to categorize the stimulus as "left" or "right." Stimulus contrast is amplified and spatial frequency reduced in this illustration. **b)** Category information is determined by the expected ratio of frames in which the orientation matches the correct category, and sensory information is determined by a parameter $\kappa$ determining the degree of spatial orientation coherence (Methods). At the start of each block, we reset the staircase to the same point, with category information at 9:1 and $\kappa$ at 0.8. We then ran a 2-to-1 staircase either on $\kappa$ or on category information. The *Low-Sensory-High-Category (LSHC)* and *High-Category-Low-Sensory (HSLC)* ovals indicate sub-threshold trials; only these trials were used in the regression to infer observers' temporal weights. **c)** Visualization of a noisy stimulus in the LSHC condition. All frames are oriented to the left. **d)** Psychometric curves for all observers (thin lines) and averaged (thick line) over the $\kappa$ staircase. Shaded gray area indicates the median threshold level across all observers. **e)** Example frames in the HSLC condition. The orientation of each frame is clear, but orientations change from frame to frame. **f)** Psychometric curves over frame ratios, plotted as in (d).

We performed additional simulations to explore the interaction between leaky integration and hierarchical inference. First, we observed that leaky integration can, surprisingly, *improve* performance, since it counteracts the confirmation bias when category information is high (Fig E in S1 Text). We further observed that optimizing the magnitude of the leak for maximum performance, separately for each combination of sensory information and category information, always resulted in flat temporal weights (Fig F in S1 Text).

Our models make a critical prediction that is not shared by any other model: that the temporal bias of the very same observer should change from primacy to flat or recency in a task in which nothing changes apart from the balance between category and sensory information.

## Changing category information in a visual discrimination task

To test this prediction, we designed an orientation discrimination task with two stimulus conditions that correspond to the two opposite sides of this task space (LSHC and HSLC), while keeping all other aspects of the design the same (Fig 4A and 4B). Importantly, in this

experiment, within-observer comparisons between the two task conditions isolate relative changes in sensory information and category information. This overcomes the difficulties in directly quantifying sensory information and category information as "high" or "low" in an isolated task, which requires additional assumptions, as discussed above.

The stimulus in our task consisted of a sequence of ten visual frames (Fig 4A). Each frame consisted of band-pass-filtered white noise with excess orientation power either in the −45˚ or the +45˚ orientation [48] (Fig 4B and 4D). On each trial, there was a single true orientation category, but individual frames might differ in their orientation. At the end of each trial, observers reported whether the stimulus was oriented predominantly in the −45˚ or the +45˚ orientation (Methods).

Sensory information in our task is determined by how well each image determines the orientation of that frame (i.e. the amount of "noise" in each frame), and category information is determined by the probability that any given frame's orientation matches the trial's category. We used signal detection theory to quantify both sensory information and category information as the area under the receiver-operating-characteristic curve for $e_f$ and $x_f$ (sensory information), and for $x_f$ and $C$ (category information). Hence for a ratio of 5 : 5 frames of each orientation, a frame's orientation does not predict the correct choice and category information is 0.5. For a ratio of 10 : 0, knowledge of the orientation of a single frame is sufficient to determine the correct choice and category information is 1. Quantifying sensory information depends on individual observer's sensory noise, but likewise ranges from 0.5 to 1 (see S1 Text).

For each observer, we compared two conditions intended to probe the difference between the LSHC and HSLC regimes. Starting with a stimulus containing both high sensory and high category information, we either ran a 2:1 staircase lowering the sensory information while keeping category information high, or we ran a 2:1 staircase lowering category information while keeping sensory information high (Fig 4B). Sub-threshold trials in each condition define the LSHC and HSLC regimes, respectively (Fig 4C and 4E). For each condition and each observer, we used logistic regression to infer the influence of each frame onto their choice. observers' overall performance was matched in the two conditions by setting a performance threshold below which trials were included in the analysis (Methods).

In agreement with our hypothesis, we find predominantly flat or decreasing temporal weights in the LSHC condition (Fig 5A), and when the information is partitioned differently—in the HSLC condition—we find flat or increasing weights (Fig 5B). To quantify this change, we first used cross-validation to select a method for quantifying temporal slopes, and found that constraining weights to be either a linear or exponential function of time worked equally well, and both outperformed logistic regression with a smoothness prior (Fig B in S1 Text; Methods). A within-observer comparison revealed that the change in slope between the two conditions was as predicted for all observers (Fig 5H) ($p < 0.05$ for 9 of 12 observers, bootstrap). The effect was also highly significant on a population level ($p < 0.01$, sign test on median slope parameters for each observer). This demonstrates that the trade-off between sensory and category information in a task robustly changes observers' temporal weighting strategy as we predicted.

## Confirmation bias, not bounded integration, explains primacy effects

The primary alternative explanation for primacy effects in fixed-duration integration tasks proposes that observers integrate evidence to an internal *bound*, at which point they cease paying attention to the stimulus [7]. In this scenario, early evidence always enters the decision-making process while evidence late in trial is often ignored. Averaged over many trials, this

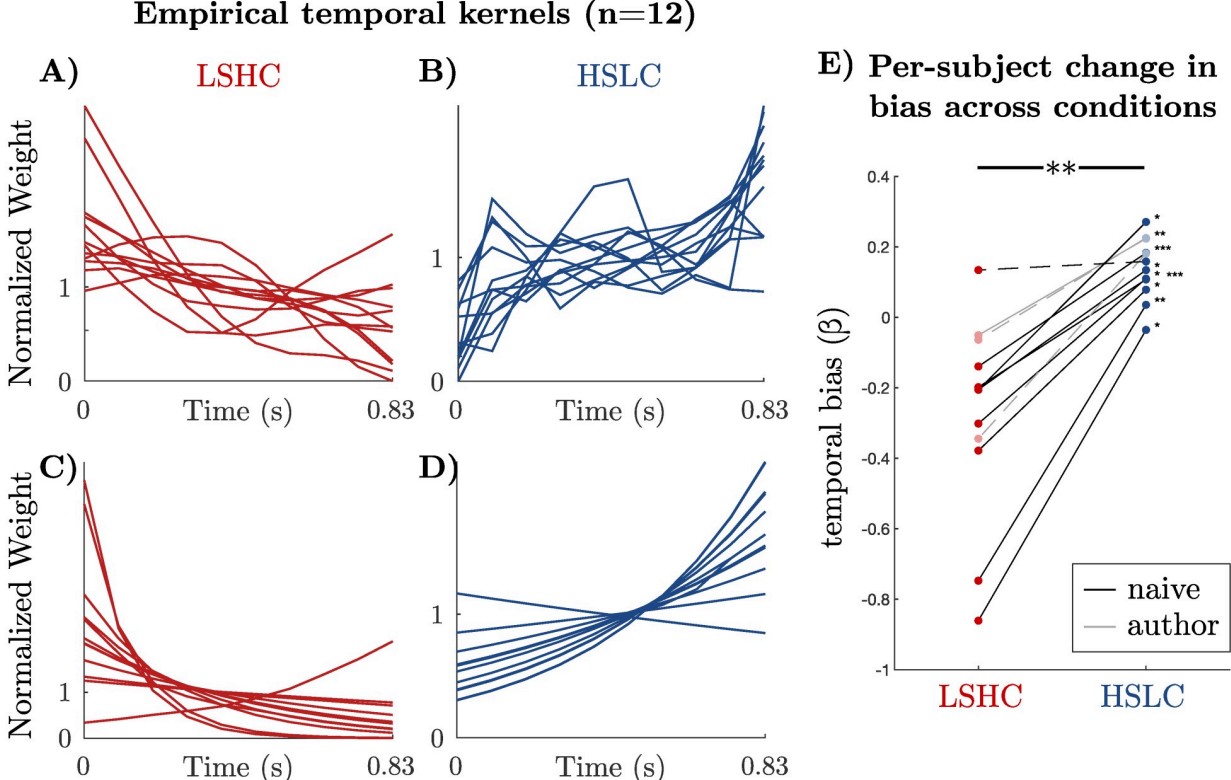

**Fig 5. Every observer's temporal bias consistently changes from primacy to unbiased/recency between conditions as predicted. a-b)** Temporal weights from logistic regression of choices from sub-threshold frames for individual observers. Weights are regularized by a cross-validated smoothness term, and are normalized to have a mean of 1. **c-d)** To summarize temporal biases, we constrained weights to be an exponential function of time and re-fit them to observers' choices. Exponential weights had higher cross-validated performance than regularized logistic regression, supporting their use to summarize temporal biases (Fig B in S1 Text; Methods). **e)** The *change* in the temporal bias, quantified as the exponential slope parameter ($\beta$), between the two task contexts for each observer is consistently positive (combined, $p < 0.01$, sign test on median slope from bootstrapping). This result is individually significant in 9 of 12 observers by bootstrapping ($p < 0.05$, $p < 0.01$, and $p < 0.001$ indicated by $*$, $**$, and $***$ respectively; non-significant observers plotted with dashed lines). Points are median slope values after bootstrap-resampling each observer's sub-threshold trials. A slope parameter $\beta > 0$ corresponds to a recency bias and $\beta < 0$ to a primacy bias. We found similar results using linear rather than exponential weight functions (Fig C in S1 Text).

results in early evidence having a larger effect on the final decision than late evidence, and hence decreasing regression weights (and psychophysical kernels) just as we found in the LSHC condition. Both models reflect very different underlying mechanism: in our approximate hierarchical inference models, a confirmation bias ensures that early evidence has a larger effect on the final decision than late evidence for every single trial. In the integration-to-bound (ITB) model, in a single trial, all evidence is weighed exactly the same before the bound is reached, and not at all afterwards.

In order to test whether the ITB mechanism or confirmation bias dynamics better explain our data, we used an Extended ITB model that can account for both biased integration dynamics (as during a confirmation bias), and for incomplete evidence accumulation due to a finite bound [49]. This model is a simple extension to classic drift diffusion models [5]. Until it hits a bound or the trial ends, the model integrates signals as follows:

$$\text{LPO}_f = (1 - \alpha)\text{LPO}_{f-1} + \text{LLO}_f + \epsilon_f \tag{2}$$

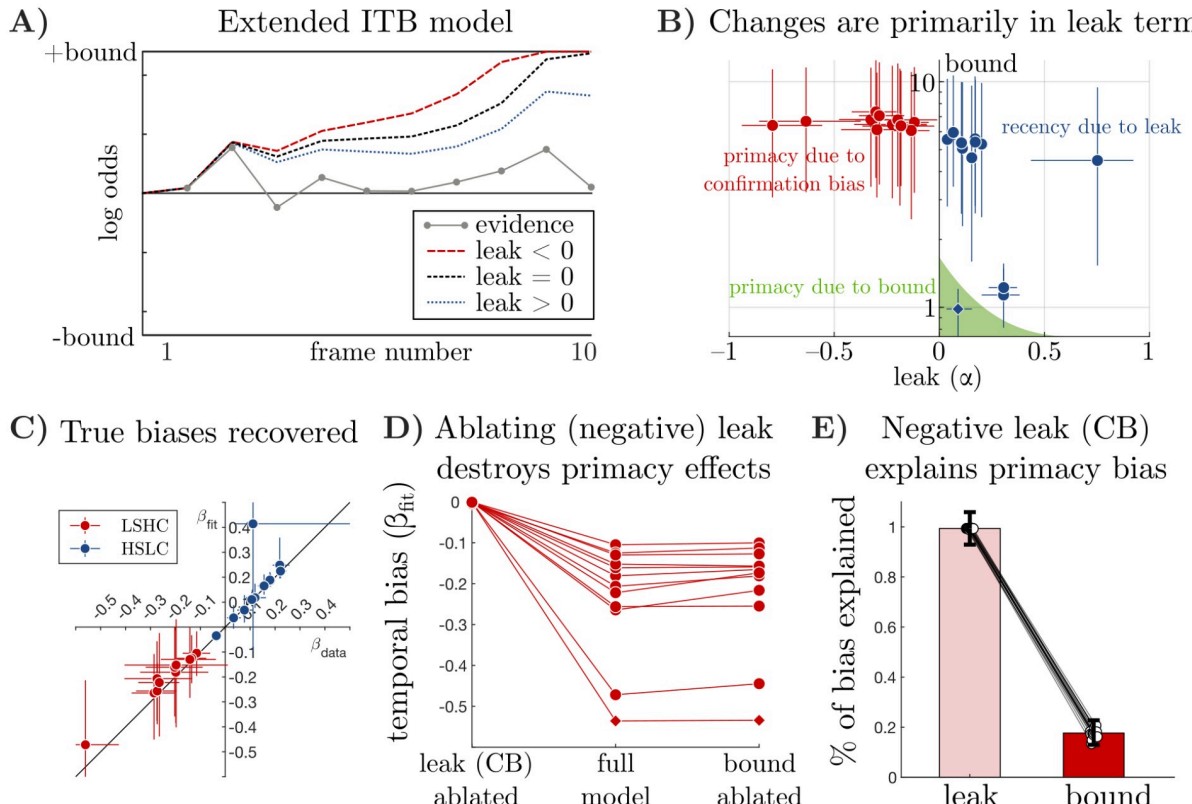

**Fig 6. Fitting an extended integration-to-bound ("Extended ITB") model to data demonstrates that integration dynamics (negative $\alpha$ for confirmation bias, positive $\alpha$ for forgetting), rather than a bound, best accounts for data. a)** Illustration of Extended ITB model. As in classic drift-diffusion models with an absorbing bound, evidence is integrated to an internal bound, after which new evidence is ignored. Compared to perfect integration ($\alpha = 0$), a *positive* leak ($\alpha > 0$) decays information away and results in a recency bias, and a negative leak ($\alpha < 0$) amplifies already integrated information, resulting in a primacy bias. Since $\alpha < 0$ may also result in more bound crossings, both leak and bound together determine the shape of the temporal weights. **b)** Inferred values of the bound and leak parameters in each condition, shown as median±68% credible intervals. The classic ITB explanation of primacy effects corresponds to a non-negative leak and a small bound—illustrated here as a shaded green area. Note that the three observers near the ITB regime are points from the HSLC task—two still exhibit mild recency effects and one exhibits a mild primacy effect as predicted by ITB. **c)** Across both conditions, the temporal slopes ($\beta$) implied by the full model fits closely match the slopes in the data. $\beta < 0$ corresponds to primacy, and $\beta > 0$ to recency. Error bars indicate 68% confidence intervals from bootstrapping trials on $\beta_{data}$ and from posterior samples on $\beta_{fit}$. **d)** Median temporal biases implied by the full model (middle) and by the model with either zero leak (left) or infinite bound (right). Each line corresponds to a single observer. (LSHC condition only—HSLC condition in Fig L in S1 Text). **d)** Across the population, the negative leak (confirmation bias) accounted for 99% (68%CI = [93%, 106%]), and bounded integration accounted for 18% (68%CI = [13%, 23%]) of the primacy bias captured by the model. (Additional analyses in Fig L in S1 Text).

where $\epsilon_f$ represents noise in the accumulation process (Fig 6A, Methods). For $\alpha = 0$, this model weighs evidence equally (optimally) over time. $\alpha > 0$ has previously been proposed to model "forgetful integration" in mechanistic models [3–5, 11], or as reflecting an assumption of non-stationarity in the environment in normative models [14], and leads to a recency effect. Importantly, a negative value for $\alpha$ induces "accelerating" integration dynamics, in which already-accumulated evidence is amplified, leading to primacy effects [3, 4, 49].

The Extended ITB model produces three distinct patterns in the data (colored text in Fig 6B). First, when $\alpha$ is positive and the bound is large, it produces recency biases. Second, when the bound is small, it produces primacy biases [7], as long as $\alpha$ is also small so that it does not prevent the bound from being crossed. Third, when the bound is large and $\alpha$ is *negative*, it also produces primacy biases but now due to confirmation-bias-like dynamics rather than due to

bounded integration. Crucially, this single model can account for both primacy due to a bound and primacy due to a confirmation bias by different parameter values (recovery of ground-truth mechanisms shown in Figs J and K in S1 Text). Examining the parameters of this model fit to data therefore allows us to determine the relative contributions of bounded integration and confirmation bias dynamics in cases where observers show primacy effects.

We fit the Extended ITB model to individual choices on sub-threshold trials, separately for the LSHC and the HSLC conditions. Fig 6B shows the posterior mean and 68% credible interval for the dynamics parameter, $\alpha$, and bound parameter inferred for each observer. The model consistently inferred a negative $\alpha$ in the LSHC condition and a positive $\alpha$ in the HSLC condition for all observers, suggesting that confirmation-bias dynamics are crucial to explain observer's primacy biases in the LSHC condition, as well as the change in bias from LSHC to HSLC conditions. Note that the leak term in the Extended ITB model reflects a combination of both the confirmation bias dynamic and a mechanistic "forgetting" term in the accumulation of the decision variable. Those two effects are opposite in nature. As a result, the acceleration inferred by our function model fit to data is likely a lower bound on the actual strength of the confirmation bias dynamics. However, while the inferred bound for every single observer is so high as not to contribute at all *if the leak was zero*, it is possible that bounded integration still contributes to primacy effects, given that a stronger confirmation bias ($\alpha < 0$) will hit a bound more often.

We therefore performed an ablation analysis to quantify the relative contribution of the leak and bound parameters to the primacy effect in the LSHC condition (Methods). We first asked whether the inferred model parameters reproduced the observed biases. Indeed, Fig 6C shows near-perfect agreement between the temporal biases implied by simulating choices from the fitted models and the biases inferred directly from observers' choices. Given this, if setting the bound to $\infty$ leaves temporal biases unchanged, then we can conclude that biases were driven by the leak, and conversely, a temporal bias that remains after setting $\alpha$ to zero must be due to the bound. Fig 6D shows that primacy effects largely disappear when $\alpha$ is ablated, but not when the bound is ablated. To summarize ablation effects across the population, we used a hierarchical regression model to compute a population-level "ablation index" for each parameter, which is 0 if removing the parameter has no effect on temporal slopes, $\beta$, and is 1 if removing it destroys all temporal biases (Methods). The ablation index can therefore be interpreted as the fraction of the observers' primacy or recency biases that are attributable to each parameter (but they do not necessarily sum to 1 because the slope is a nonlinear combination of both parameters). In the LSHC condition, the ablation index for the leak term was 0.99 (68% CI = [0.93, 1.06]), and for the bound term it was 0.18 (68% CI = [0.13, 0.23]) (Fig 6E). This indicates that although both mechanisms are present, primacy effects in our data are dominated by the self-reinforcing dynamics of a negative leak. Results for the HSLC condition are shown in Fig L in S1 Text.

Interestingly, one observer exhibited a slight primacy effect in the HSLC condition, and our analyses suggest this was primarily due to bounded integration dynamics as proposed by Kiani et al (2008). This outlier observer is marked with a diamond symbol throughout Fig 6, and is further highlighted in Fig L in S1 Text. However, even this observer's primacy effect in the LSHC condition was driven by a confirmation bias (negative $\alpha$), and their change in slope between LSHC and HSLC conditions was in the same direction as the other observers. Importantly, finding a primacy effect due to an internal bound confirms that our model fitting procedure is able to detect such effects when they are, in fact, present. Two additional observers appear to have low bounds in the HSLC condition (Fig 6C), but are dominated by leaky integration ($\alpha > 0$), resulting in an overall recency bias.

## Discussion

Our work makes three main contributions. First, we extended ideal observer models of evidence integration tasks by explicitly accounting for the intermediate sensory representation. We showed that this partitions the information in the stimulus about the category into two parts—"sensory information" and "category information"—defining a novel two-dimensional space of possible tasks. Second, we found that two classes of biologically-plausible approximate inference algorithms entailed a confirmation bias whose strength strongly varied across this task space. Interestingly, the location of tasks in existing studies qualitatively predicted the bias they found across species, sensory modalities and task designs. Third, we collected new data and confirmed a critical prediction of our theory, namely that individual observers' temporal biases should change depending on the balance of sensory information and category information in the stimulus. Finally, by fitting an extended integration to bound (Extended ITB) model to individual observer choices, we confirmed that these changes in biases are due to a change in integration dynamics rather than bounded integration.

The "confirmation bias" emerges in our hierarchical inference models as the result of three key assumptions. Our first assumption is that inference in evidence integration tasks is in fact hierarchical, and that the brain approximates the posterior distribution over the intermediate sensory variables, $x$. This is in line with converging evidence that populations of sensory neurons encode posterior distributions of corresponding sensory variables [34, 35, 50, 51] incorporating dynamic prior beliefs via feedback connections [34, 35, 39, 42, 43, 51–53]. This is in contrast to other probabilistic theories in which only the likelihood is represented in sensory areas [25, 26, 28, 54], which would not predict primacy effects due to confirmation bias dynamics.

Our second key assumption is that evidence is accumulated online. In our models, the belief over $C$ is updated based only on the posterior from the previous step and the current posterior over $x$. This can be thought of as an assumption that the brain does not have a mechanism to store and retrieve earlier frames of evidence directly, and is consistent with drift-diffusion models of decision-making [5]. As mentioned in the main text, the assumptions until now—hierarchical inference with online updates—do not entail any temporal biases for an ideal observer. Further, the use of discrete time in our experiment and models is only for mathematical convenience—analogous dynamics emerge in continuous-time, and in fact we implemented our models at a finer time scale than at which evidence frames are presented.

Third, we assumed that inference in the brain is approximate—a safe assumption due to the intractable nature of exact inference in large models. In the sampling model, we assumed that the brain can draw a limited number of independent samples of $x$ per update to $C$, and found that for a finite number of samples the model is inherently unable to account for all of the prior bias of $C$ on $x$ in its updates to $C$. Existing neural models of sampling typically assume that samples are distributed temporally [36, 42, 53, 55, 56], but it has also been proposed that the brain could process multiple sampling "chains" distributed spatially [57]. The relevant quantity for our model is the number of independent samples that can be tallied per update: the more samples, the smaller the bias. The variational model's representational capacity was limited by enforcing that the posterior over $x$ is unimodal, and that there is no explicit representation of dependencies between $x$ and $C$. Importantly, this does not imply that $x$ and $C$ do not influence each other. Rather, the Variational Bayes algorithm expresses these dependencies in the *dynamics* between the two areas: each update that makes $C = +1$ more likely pushes the distribution over $x$ further towards +1, and vice versa. Because the number of dependencies between variables grows exponentially, such approximates are necessary in variational

inference with many variables [36]. The Mean Field Variational Bayes algorithm that we use here has been previously proposed as a candidate algorithm for neural inference [58].

The assumptions up to now predict a primacy effect but cannot account for the observed recency effects. When we incorporate a forgetting term in our models, they reproduce the observed range of biases from primacy to recency. The existence of such a forgetting term is supported by previous literature [4, 15]. Further, it is normative in our framework in the sense that reducing the bias in the above models improves performance (Fig D in S1 Text through Fig F in S1 Text). The optimal amount of bias correction depends on the task statistics: for high category information where the confirmation bias is strongest, a stronger forgetting term is needed to correct for it. While it is conceivable that the brain would optimize this term to the task [11, 59, 60], our data suggest it is stable across our LSHC and HSLC conditions, or only adapts slowly.

It has been proposed that post-decision feedback biases subsequent perceptual estimations [61–65]. While in spirit similar to our confirmation bias model, there are two conceptual differences between these models and our own: First, the feedback from decision area to sensory area in our model is both continuous and online, rather than conditioned on a single choice after a decision is made. Second, our models are derived from an ideal observer and only incur bias due to algorithmic approximations, while previously proposed "self-consistency" biases are not normative and require separate justification. However, these previous findings on the effect of commitment to a choice on weighing subsequent evidence can easily be accommodated in our framework by plausibly proposing that the act of committing to a choice increases one's subjective certainty about that choice being correct. In our model, such an increase in certainty would directly translate into an increase in feedback from decision to sensory areas, and hence increased confirmation bias.

Our analysis decisively shows that accelerating dynamics, rather than reaching a bound before the end of the trial, explains the primacy effect in our data. Prior work has suggested that such accelerating dynamics may arise from mutual inhibition of two accumulators [15, 18, 20], or two attractor states corresponding to the two choices [16, 19, 66–68]—all *within* a decision-area, and that the nature of the temporal bias depends on the volatility of the integrated signal [22]. Importantly, decision-dynamics alone *cannot* explain our results, since the input to these models usually reflects the total information in each frame about the choice, i.e. combining both sensory and category information. In other words, these models usually integrate *log odds*, which we kept approximately constant between LSHC and HSLC conditions. The same argument applies to other models that do no distinguish between sensory and category information, whether based on mixing trials of different difficulty [21] or differential accumulation of consistent and inconsistent evidence [63, 64, 69, 70].

In contrast, in our explanation based on approximate hierarchical inference, attractor dynamics arise *across* sensory and decision areas, as the result of cortical inter-area feedback whose strength is monotonically related to category information. Holding the task difficulty (and hence the magnitude of the log odds of each frame) constant, our model nonetheless predicts stronger inter-area attractor dynamics when category information is high. Given recent evidence that noise correlations contain a task-dependent feedback component [71], we therefore predict a reduction of task-dependent noise correlations in comparable tasks with lower category information. The confirmation bias mechanism may also account for the recent finding that stronger attractor dynamics are seen in a categorization task than in a comparable estimation task [38].

Rollwage et al (2020) recently presented evidence for a relationship between decision confidence and confirmation bias: when observers are more confident about a decision, they will be more biased in how they interpret subsequent information [65]. Interestingly, our model

makes a closely related prediction: that positive feedback loop between decision and sensory area also increases confidence beyond that of an ideal observer. In particular, it predicts that confidence judgements in the LSHC condition when the primacy effect is strong will be higher than in the HSLC condition when the primacy effect is weak—a prediction that we found to be confirmed in follow-up work [72].

It has also been proposed that primacy effects could be the result of near-perfect integration of an adapting sensory population [29, 68]. For this mechanism to explain our full results, however, the sensory population would need to become *less* adapted over the course of a trial in our HSLC condition, while at the same time *more* adapted in the LSHC condition. We are unaware of such an adaptation mechanism in the literature. Further, such stimulus-dependent circuit dynamics would not predict top-down neural effects such as the task-dependence of the dynamics of sensory populations [38] nor the origin and prevalence of differential correlations [71], both of which are consistent with hierarchical inference [39, 42].

While our focus is on the perceptual domain in which observers integrate evidence over a timescale on the order of tens or hundreds of milliseconds, analogous computational principles hold in the cognitive domain over longer timescales. The crucial computational motif underlying our model of the confirmation bias is approximate hierarchical inference over multiple timescales. An agent in such a setting must simultaneously make accurate judgments of current data (based on the current posterior) and track long-term trends (based on all likelihoods). For instance, Zylberberg et al. (2018) identified an analogous challenge when observers must simultaneously make categorical decisions each trial (their "fast" timescale) while tracking the stationary statistics of a block of trials (their "slow" timescale), with trial-by-trial statistics analogous to the frame-by-frame statistics in our LSHC condition. As the authors describe, if observers base model updates on posteriors rather than likelihoods, they will further entrench existing beliefs [73]. However, the authors did not investigate order effects; our proposed confirmation bias models would predict that observers' estimates of block statistics is biased towards earlier trials in the block (primacy). Schustek et al. (2018) likewise asked observers to track information across trials in a cognitive task more analogous to our HSLC condition, and report close to flat weighting of evidence across trials [74] in agreement with our model.

The strength of the perceptual confirmation bias is directly related to the integration of internal "top-down" beliefs and external "bottom-up" evidence previously implicated in clinical dysfunctions of perception [75]. Therefore, the differential effect of sensory and category information may be useful in diagnosing clinical conditions that have been hypothesized to be related to abnormal integration of sensory information with internal expectations [76].

Hierarchical (approximate) inference on multiple timescales is a common motif across perception, cognition, and machine learning. We suspect that all of these areas will benefit from the insights on the causes of the confirmation bias mechanism that we have described here and how they depend on the statistics of the inputs in a task.

## Methods

### Ethics statement

This study was approved by the Institutional Research Review Board of the University of Rochester (RSRB #55456).

### Visual discrimination task

We recruited twelve students at the University of Rochester as observers in our study. All non-author participants were compensated for their time. We found no difference between naive

observers and authors, so all main-text analyses are combined, with data points belonging to authors and naive observers indicated in Fig 5D.

Our stimulus consisted of ten frames of band-pass filtered noise [48, 77] masked by a soft-edged annulus, leaving a "hole" in the center for a small cross on which observers fixated. The stimulus subtended 2.6 degrees of visual angle around fixation. Stimuli were presented using Matlab and Psychtoolbox on a 1920x1080px 120 Hz monitor with gamma-corrected luminance [78]. Observers kept a constant viewing distance of 36 inches using a chin-rest. Each trial began with a 200ms "start" cue consisting of a black ring around the location of the upcoming stimulus. Each frame lasted 83.3ms (12 frames per second). The last frame was followed by a single double-contrast noise mask with no orientation energy. Observers then had a maximum of 1s to respond, or the trial was discarded (Fig 4A). The stimulus was designed to minimize the effects of small fixational eye movements: (i) small eye movements do not provide more information about either orientation, and (ii) each 83ms frame was too fast for observers to make multiple fixations on a single frame.

The stimulus was constructed from white noise that was then masked by a kernel in the Fourier domain to include energy at a range of orientations and spatial frequencies but random phases [48, 71, 77] (a complete description and parameters can be found in Table B in S1 Text). We manipulated sensory information by broadening or narrowing the distribution of orientations present in each frame, centered on either +45° or −45° depending on the chosen orientation of each frame. We manipulated category information by changing the proportion of frames that matched the orientation chosen for that trial. The range of spatial frequencies was kept constant for all observers and in all conditions.

Trials were presented in blocks of 100, with typically 8 blocks per session (about 1 hour). Each session consisted of blocks of only HSLC or only LSHC trials (Fig 4). Observers completed between 1500 and 4400 trials in the LSHC condition, and between 1500 and 3200 trials in the HSLC condition. After each block, observers were given an optional break and the staircase was reset to $\kappa = 0.8$ and $p_{match} = 0.9$. $p_{match}$ is defined as the probability that a single frame matched the category for a given trial. In each condition, psychometric curves were fit to the concatenation of all trials from all sessions using the Psignifit Matlab package [79], and temporal weights were fit to all trials below each observer's threshold.

**Low sensory-, high category-information (LSHC) condition.**   In the LSHC condition, a continuous 2-to-1 staircase on $\kappa$ was used to keep observers near threshold ($\kappa$ was incremented after each incorrect response, and decremented after two correct responses in a row). $p_{match}$ was fixed to 0.9. On average, observers had a threshold (defined as 70% correct) of $\kappa = 0.17 \pm 0.07$ (1 standard deviation). Regression of temporal weights was done on all sub-threshold trials, defined per-observer.

**High sensory-, low category-information (HSLC) condition.**   In the HSLC condition, the staircase acted on $p_{match}$ while keeping $\kappa$ fixed at 0.8. Although $p_{match}$ is a continuous parameter, observers always saw 10 discrete frames, hence the true ratio of frames ranged from 5:5 to 10:0 on any given trial. Observers were on average 69.5% ± 4.7% (1 standard deviation) correct when the ratio of frame types was 6:4, after adjusting for individual biases in the 5:5 case. Regression of temporal weights was done on all 6:4 and 5:5 ratio trials for all observers, regardless of the underlying $p_{match}$ parameter.

## Logistic regression of temporal weights

We constructed a matrix of per-frame signal strengths **S** on sub-threshold trials by measuring the empirical signal level in each frame. This was done by taking the dot product of the Fourier-domain energy of each frame as it was displayed on the screen (that is, including the

annulus mask applied in pixel space) with a difference of Fourier-domain kernels at +45˚ and −45˚ with $\kappa = 0.16$. This gives a scalar value per frame that is positive when the stimulus contained more +45˚ energy and negative when it contained more −45˚ energy. Signals were z-scored before performing logistic regression, and weights were normalized to have a mean of 1 after fitting.

Temporal weights were first fit using (regularized) logistic regression with different types of regularization. The first regularization method consisted of an AR0 (ridge) prior, and an AR2 (curvature penalty) prior. We did not use an AR1 prior to avoid any bias in the slopes, which is central to our analysis.

To visualize regularized weights in Fig 5, the ridge and AR2 hyperparameters were chosen using 10-fold cross-validation for each observer, then averaging the optimal hyperparameters across observers for each task condition. This cross validation procedure was used only for display purposes for individual observers in Fig 5A and 5B of the main text, while the linear and exponential fits (described below) were used for Fig 5C and 5D and statistical comparisons. Fig A in S1 Text shows individual observers' weights for all regression models.

We used two methods to quantify the shape (or slope) of $\mathbf{w}$: by constraining $\mathbf{w}$ to be either an exponential or linear function of time, but otherwise optimizing the same maximum-likelihood objective as logistic regression. Cross-validation suggests that both of these methods perform similarly to either unregularized or the regularized logistic regression defined above, with insignificant differences (Fig B in S1 Text). The exponential is defined as

$$\mathbf{w}_f^{\text{exponential}} = \alpha \, \exp \, (\beta f) \tag{3}$$

where $f$ refers to the frame number. $\beta$ gives an estimate of the shape of the weights $\mathbf{w}$ over time, while $\alpha$ controls the overall magnitude. $\beta > 0$ corresponds to recency and $\beta < 0$ to primacy. The $\beta$ parameter is reported for human observers in Fig 5E, and for the models in Fig 3D and 3G.

The second method to quantify slope was to constrain the weights to be a linear function in time:

$$\mathbf{w}_f^{\text{linear}} = a + slope \times f \tag{4}$$

where $slope > 0$ corresponds to recency and $slope < 0$ to primacy.

Fig 5E shows the median exponential shape parameter ($\beta$) after bootstrapped resampling of trials 500 times for each observer. Both the exponential and linear weights give comparable results (Fig C in S1 Text).

Because we are not explicitly interested in the magnitude of $\mathbf{w}$ but rather its *shape* over stimulus frames, we always plot a "normalized" weight, $\mathbf{w}/\text{mean}(\mathbf{w})$, both for our experimental results (Fig 5A–5D) and for the model (Fig 3C and 3F).

## Approximate inference models

We model evidence integration as Bayesian inference in a three-variable generative model (Fig 2A) that distills the key features of online evidence integration in a hierarchical model [42]. The variables in the model are mapped onto the sensory periphery ($e$), sensory cortex ($x$), and a decision-making area ($C$) in the brain. For simulations, the same model was used both to generate data ($C \rightarrow x_f \rightarrow e_f$), and, in the reverse direction, as a model of inference dynamics ($e_f \rightarrow \text{p}(x_f \,|\, \ldots) \leftrightarrow \text{p}(C \,|\, \ldots)$).

In the generative direction, on each trial, the binary value of the correct choice $C \in \{-1, +1\}$ is drawn from a 50/50 prior. $x_f$ is then drawn from a mixture of two Gaussians:

$$x_f^{(gen)} \sim \begin{cases} \mathcal{N}(+C, \sigma_x^2) & \text{with prob. equal to category info.} \\ \mathcal{N}(-C, \sigma_x^2) & \text{otherwise} \end{cases} \quad (5)$$

Finally, each $e_f$ is drawn from a Gaussian around $x_f$:

$$e_f^{(gen)} \sim \mathcal{N}(x_f, \sigma_e^2) . \quad (6)$$

In the inference direction, we assume that the observer has learned the correct model parameters (namely the category information, and sensory information or $\sigma_e^2$), even as parameters change between the two different conditions. This is why we ran our observers in blocks of only LSHC or HSLC trials on a given day.

Category information in this model can be quantified by the probability that $x_f^{(gen)}$ is drawn from the mode that matches $C$, as in Eq (5). We quantify sensory information as the probability with which an ideal observer can recover the sign of $x_f$ from a single $e_f$. That is, in our model sensory information is equivalent to the area under the ROC curve for two univariate Gaussian distributions separated by a distance of 2, which is given by

$$\text{sensory info.} = \Phi(\sqrt{2}/\sigma_e) \quad (7)$$

where $\Phi$ is the inverse cumulative normal distribution.

Optimal inference of $C$ requires conditioning on all frames of evidence $e_1, \ldots, e_f$, which can be expressed as the Log Posterior Odds (LPO),

$$\underbrace{\log \frac{p(C = +1 | e_1, \ldots, e_f)}{p(C = -1 | e_1, \ldots, e_f)}}_{\text{LPO}_f} = \log \frac{p(C = +1)}{p(C = -1)} + \sum_{i=1}^{f} \underbrace{\log \frac{p(e_i | C = +1)}{p(e_i | C = -1)}}_{\text{LLO}_i}, \quad (8)$$

where $\text{LLO}_f$ is the log likelihood odds for frame $f$ [4, 5]. To reflect the fact that the brain has access to only one frame of evidence at a time, this can be rewritten this as an *online* update rule, summing the previous frame's log posterior with new evidence gleaned on the current frame:

$$\text{LPO}_f = \text{LPO}_{f-1} + \text{LLO}_f. \quad (9)$$

Optimal inference of $x_f$ similarly requires accounting for all possible sources of information. Ideally, sensory areas should incorporate prior information based on previous frames to compute $p(x_f | e_1, \ldots, e_f)$. Using $p_{f-1}(C = c) = p(C = c | e_1, \ldots, e_{f-1})$ to denote the brain's belief that the category is $C = c$ after the first $f - 1$ frames, the posterior over $x_f$ given *all* frames, $p(x_f | e_1, \ldots, e_f)$, can be written as

$$p(x_f | e_1, \ldots, e_f) \propto p(e_f | x_f) \underbrace{\sum_c p_{f-1}(C = c) p(x_f | C = c)}_{p_f(x_f)} . \quad (10)$$

The term $p_f(x_f)$ is a prior on sensory features $x_f$ that changes over time depending on the current belief in the category, $p_{f-1}(C)$. In other words, sensory areas could dynamically combine instantaneous evidence ($p(e_f | x_f)$) with accumulated categorical beliefs ($p_{f-1}(C)$) to arrive at a more precise estimate of present sensory features $x_f$. This is what we mean by feedback of "decision-related information" or of "categorical beliefs."

Our approximate inference models, described in detail below, compute a *biased* estimate of $\text{LLO}_f$, which we call $\text{L}\hat{\text{L}}\text{O}_f$. The bias is due to the interaction of approximations with feedback of prior beliefs, such that $L\hat{L}O_f$ is biased towards $\text{LPO}_{f-1}$, resulting in a confirmation bias. Importantly, this bias arises naturally in both the sampling-based and variational approximate inference algorithms that we study here, as a direct consequence the *approximate* nature of the posterior. Given the approximate representation of posteriors over $x_f$, there is no way to exactly divide out the influence of the prior and recover the exact log likelihood odds on a frame-by-frame basis. However, the confirmation bias can be mitigated on average simply by incorporating a leak term, $\gamma$, in the integration process [3, 4]:

$$\text{LPO}_f \leftarrow (1 - \gamma)\text{LPO}_{f-1} + \text{L}\hat{\text{L}}\text{O}_f \,. \tag{11}$$

Due to the bias in $\text{L}\hat{\text{L}}\text{O}_f$, $\gamma$ can be seen as a kind of approximate *bias correction*, with positive values for $\gamma$ often improving performance (Fig E in S1 Text through Fig F in S1 Text). Equivalently, one can view the quantity $\text{L}\hat{\text{L}}\text{O}_f - \gamma\text{LPO}_{f-1}$ as a less biased estimate of the true log likelihood odds.

Because the effective time per update in the brain is likely faster than our 83ms stimulus frames, we included an additional parameter $n_U$ for the number of online belief updates per stimulus frame. In the sampling model described below, we amortize the per-frame updates over $n_U$ steps, updating $n_U$ times per frame using $\frac{1}{n_U}\text{L}\hat{\text{L}}\text{O}_f$. In the variational model, we interpret $n_U$ as the number of coordinate ascent steps per stimulus frame.

Simulations of both models were done with 10000 trials per task type and 10 frames per trial. To quantify the evidence-weighting of each model, we used the same logistic regression procedure that was used to analyze human observers' behavior. In particular, temporal weights in the model are best described by the exponential weights (Eq (3)), so we use $\beta$ to characterize the model's biases.

**Sampling model.** The sampling model estimates $p(e_f|C)$ using importance sampling of $x$, where each sample is drawn from a pseudo-posterior using the current running estimate of $p_{f-1}(C) \equiv p(C|e_1, .., e_{f-1})$ as a marginal prior:

$$x_f^{(s)} \sim Q(x) \propto p(e_f|x_f)\sum_c p(x_f|C = c)p_{f-1}(C = c) \tag{12}$$

Using this distribution, we obtain the following unnormalized importance weights.

$$w^{(s)} = \left(\sum_c p(x_f^{(s)}|C = c)p_{f-1}(C = c)\right)^{-1} \tag{13}$$

In the self-normalized importance sampling algorithm these weights are then normalized as follows,

$$\hat{w}^{(s)} = \frac{w^{(s)}}{\sum_i w^{(i)}} \,,$$

though we found that this had no qualitative effect on the model's ability to reproduce the trends in the data. The above equations yield the following estimate for the log-likelihood ratio

needed for the belief update rule in Eq (11):

$$\hat{\text{LLO}}_f = \log \frac{\sum_{s=1}^{S} \text{p}(x_f^{(s)}|C = +1)w^{(s)}}{\sum_{s=1}^{S} \text{p}(x_f^{(s)}|C = -1)w^{(s)}} \tag{14}$$

In the case of infinitely many samples, these importance weights exactly counteract the bias introduced by sampling from the posterior rather than likelihood, thereby avoiding any double-counting of the prior, and hence, any confirmation bias [80]. However, in the case of finite samples, $S$, biased estimates of $\text{LLO}_f$ are unavoidable [81].

The full sampling model is given in Algorithm A in S1 Text. Simulations in the main text were done with $S = 5$, $n_U = 5$, normalized importance weights, and $\gamma = 0$ or $\gamma = 0.1$.

**Variational model.** The following variational model produces qualitatively similar patterns of temporal biases to the IS model (Fig D in S1 Text).

The core assumption of the variational model is that while a decision area approximates the posterior over $C$ and a sensory area approximates the posterior over $x$, no brain area explicitly represents posterior dependencies between them. That is, we assume the brain employs a *mean field approximation* to the joint posterior by factorizing $\text{p}(C, x_1, \ldots, x_F | e_1, \ldots, e_F)$ into a product of approximate marginal distributions $\text{q}(C) \prod_{f=1}^{F} \text{q}(x_f)$ and minimizes the Kullback-Leibler divergence between q and p using a process that can be modeled by the Mean-Field Variational Bayes algorithm [46].

By restricting the updates to be online (one frame at a time, in order), this model can be seen as an instance of "Streaming Variational Bayes" [82]. That is, the model computes a sequence of approximate posteriors over $C$ using the same update rule for each frame. We thus only need to derive the update rules for a single frame and a given prior over $C$; this is extended to multiple frames by re-using the posterior from frame $f - 1$ as the prior on frame $f$.

As in the sampling model, this model is unable to completely discount the added prior over $x$. Intuitively, since the mean-field assumption removes explicit correlations between $x$ and $C$, the model is forced to commit to a marginal posterior in favor of $C = +1$ or $C = -1$ and $x > 0$ or $x < 0$ after each update, which then biases subsequent judgments of each.

To keep conditional distributions in the exponential family (which is only a matter of mathematical convenience and has no effect on the ideal observer), we introduce an auxiliary variable $z_f \in \{-1, +1\}$ that selects which of the two modes $x_f$ is in:

$$z_f = \begin{cases} +1 & \text{with probability equal to category info} \\ -1 & \text{otherwise} \end{cases} \tag{15}$$

such that

$$x_f \sim \mathcal{N}(z_f C, \sigma_x^2). \tag{16}$$

We then optimize $\text{q}(C) \prod_{f=1}^{F} \text{q}(x_f)\text{q}(z_f)$.

Mean-Field Variational Bayes is a coordinate ascent algorithm on the parameters of each approximate marginal distribution. To derive the update equations for each step, we begin

with the following [46]:

$$\log q(x_f) \leftarrow \mathbf{E}_{q(C)q(z_f)}[\log p(C, x_f, z_f|e_f)] + const$$

$$\log q(z_f) \leftarrow \mathbf{E}_{q(C)q(x_f)}[\log p(C, x_f, z_f|e_f)] + const \qquad (17)$$

$$\log q(C) \leftarrow \mathbf{E}_{q(x_f)q(z_f)}[\log p(C, x_f, z_f|e_f)] + const$$

After simplifying, the new $q(x_f)$ term is a Gaussian with mean given by Eq (18) and constant variance

$$\mu_{x_f} \leftarrow \frac{\sigma_e^2 \mu_C \mu_{z_f} + \sigma_x^2 e_f}{\sigma_e^2 + \sigma_x^2} \qquad (18)$$

where $\mu_C$ and $\mu_z$ are the means of the current estimates of $q(C)$ and $q(z)$.

For the update to $q(z_f)$ in terms of log odds of $z_f$ we obtain:

$$\mathrm{LPO}_{z_f} \leftarrow \log \frac{p(z_f = +1)}{p(z_f = -1)} + 2\frac{\mu_{x_f}\mu_C}{\sigma_e^2 + \sigma_x^2}. \qquad (19)$$

Similarly, the update to $q(C)$ is given by:

$$\mathrm{LPO}_C \leftarrow \log \frac{p(C = +1)}{p(C = -1)} + 2\frac{\mu_{x_f}\mu_{z_f}}{\sigma_x^2} \qquad (20)$$

Note that the first term in Eq (20)—the log prior—will be replaced with the log posterior estimate from the previous frame (see Algorithm B in S1 Text). Comparing Eqs (20) and (9), we see that in the variational model, the log likelihood odds estimate is given by

$$\hat{\mathrm{LLO}}_f = 2\frac{\mu_{x_f}\mu_{z_f}}{\sigma_x^2} \qquad (21)$$

Analogously to the sampling model we assume a number of updates $n_U$ reflecting the speed of relevant computations in the brain relative to how quickly stimulus frames are presented. Unlike for the sampling model, naively amortizing the updates implied by Eq (21) $n_U$ times results in a stronger primacy effect than observed in the data, since the Variational Bayes algorithm naturally has attractor dynamics built in. Allowing for an additional parameter $\eta$ scaling this update (corresponding to the step size in Stochastic Variational Inference [83]) seems biologically plausible because it simply corresponds to a coupling strength in the feed-forward direction. Decreasing $\eta$ both reduces the primacy effect and improves the model's performance. Here we used $\eta = 0.05$ in all simulations based on a qualitative match with the data. The full variational model is given in Algorithm B in S1 Text.

**Fitting the extended ITB model to data.** To explore alternatives, we implemented an Integration to Bound (ITB) model in our simplified 3-variable hierarchical task model, $C \rightarrow x_f \rightarrow e_f$. The dynamics of the integrator model were nearly identical to Eq (11), using the exact log likelihood odds, but with added noise:

$$\mathrm{LPO}_f = \mathrm{LPO}_{f-1}(1 - \alpha) + \mathrm{LLO}_f + \epsilon \quad , \qquad (22)$$

where $\epsilon$ is zero-mean Gaussian noise with variance $\sigma_\epsilon^2$ [4, 11, 13, 15, 19]. Although $\alpha$ plays a similar role to $\gamma$ from the hierarchical inference models (both $\alpha$ and $\gamma$ are referred to as "leak" parameters), we distinguish between them to avoid confusion. Whereas $\alpha < 0$ produces confirmation-bias dynamics in the Extended ITB model, in the hierarchical inference models a

confirmation bias occurs when $\gamma$ is small but positive and category information is high (that is, the confirmation bias in hierarchical inference is due to biased estimation of $\hat{\text{LLO}}$ rather than to $\gamma$). In the Extended ITB model, whenever $\text{LPO}_f$ crosses the bound at $\pm B$, it "sticks" to that bound for the rest of the trial regardless of further evidence. Note that in the unbounded case noise does not affect the shape of the temporal weights (only their magnitude), but noise interacts with the bound to determine the shape as well as overall performance. Fig G in S1 Text shows the performance and temporal biases of the ITB model for a range of parameter values.

Per observer per condition, we used Metropolis Hastings (MH) to infer the joint posterior over seven parameters: the category prior ($p_C$), lapse rate ($\lambda$), decision temperature ($T$), integration noise ($\epsilon$), bound ($B$), leak ($\alpha$), and evidence scale ($s$). One challenge for fitting models is that the mapping from signal in the images ($\mathbf{S}$) to "log odds" to be integrated (LLO) depends on category information, sensory information, and on unknown properties of each observer's visual system. The evidence scale parameter, $s$, was introduced because although we can estimate the ground truth category information in each task (0.6 for HSLC and 0.9 for LSHC), the *effective* sensory information depends on each observer's visual system and will differ between the two tasks. Using logistic regression, we explored plausible nonlinear monotonic mappings between signals $\mathbf{S}$ and log-odds, but found that none performed better than linear scaling when applied to sub-threshold trials. We therefore used $\text{LLO} \approx g(\mathbf{S}/s)$, where $g$ is a sigmoidal function that accounts for category information being less than 1, and inferred $s$ jointly along with other parameters of the model. The scale $s$ was fixed to 1 when fitting the ground-truth models, as the mapping between evidence and log odds is completely known in those cases.

Each trial, the Extended ITB model followed the noisy integration dynamics in Eq (22), where $\text{LPO}_0 = \log \frac{p_C}{1-p_C}$ and $\text{LLO}_f$ was computed exactly, as described above. After integration, the decision then incorporated a symmetric lapse rate and temperature:

$$\text{p}(\text{Choice} = +1 | \text{LPO}_F, \lambda, T) = \lambda + (1 - 2\lambda)\sigma \left(\text{LPO}_F/T\right),$$

where $\sigma(a)$ is the sigmoid function, $\sigma(a) \equiv (1 + \exp(-a))^{-1}$. Note that if the bound is hit, then $\text{LPO}_F = \pm B$, but the temperature and lapse still apply. To compute the log likelihood for each set of parameters, we numerically marginalized over the noise, $\epsilon$, by discretizing LPO into bins of width at most 0.01 between $-B$ and $+B$ (clipped at 3 times the largest LPO reached by the ideal observer) and computing the *probability mass* of $\text{LPO}_f$ given $\text{LPO}_{f-1}$, $\text{LLO}_f$, and $\epsilon$. This enabled exact rather than stochastic likelihood evaluations within MH.

The priors over each parameter were set as follows. $\text{p}(p_C)$ was set to Beta(2, 2). $\text{p}(\lambda)$ was set to Beta(1, 10). $\text{p}(\alpha)$ was uniform in $[-1, 1]$. $\text{p}(s)$ was set to an exponential distribution with mean 20. $\text{p}(\epsilon)$ was set to an exponential distribution with mean 0.25. $\text{p}(T)$ was set to an exponential distribution with mean 4. $\text{p}(B)$ was set to a Gamma distribution with (shape,scale) parameters (2, 3) (mean 6). MH proposal distributions were chosen to minimize the autocorrelation time when sampling each parameter in isolation.

We ran 12 MCMC chains per observer per condition. The initial point for each chain was selected as the best point among 500 quasi-random samples from the prior. Chains were run for variable durations based on available shared computing resources. Each was initially run for 4 days; all chains were then extended for each model that had not yet converged according to the Gelman-Rubin statistic, $\hat{R}$ [84, 85]. We discarded burn-in samples separately per chain post-hoc, defining burn-in as the time until the first sample surpassed the median posterior probability for that chain (maximum 20%, median 0.46%, minimum 0.1% of the chain length for all chains). After discarding burn-in, all chains had a minimum of 81k, median 334k, and maximum 999k samples. Standard practice suggests that $\hat{R} < 1.1$ indicates good enough convergence. The slowest-mixing parameter was the signal scale ($s$), with $\hat{R} = 1.13$ in the worst

case. All $\hat{R}$ values for the parameters relevant to the main analysis—$\alpha$, $B$, and $\beta$—indicated convergence ([min, median, max] values of $\hat{R}$ equal to [1, 1.00335, 1.032] for $\alpha$, [1.0005, 1.00555, 1.0425] for $B$, and [1, 1.0014, 1.0178] for all $\beta$ values in ablation analyses.

**Estimating temporal slopes and ablation indices implied by model samples.** To estimate the the shape of temporal weights implied by the model fits, we simulated choices from the model once for each posterior sample after thinning to 500 samples per chain for a total of 6k samples per observer and condition. We then fit the slope of the exponential weight function, $\beta$, to these simulated choices using logistic regression constrained to be an exponential function of time as described earlier (Eq (3)). This is the $\beta_{\text{fit}}$ plotted on the y-axis of Fig 6C. For the ablation analyses, we again fit $\beta$ to choices simulated once per posterior sample of model parameters, but setting $\alpha = 0$ in one case or ($B = \infty$, $\epsilon = 0$) in the other.

We used a hierarchical regression analysis to compute "ablation indices" per observer and per parameter. The motivation for this analysis is that observers have different magnitudes of primacy and recency effects, but the *relative* impact of the leak or bound and noise parameters appeared fairly consistent throughout the population (Fig L in S1 Text), so a good summary index measures the *fraction* of the bias attributable to each parameter, which directly relates to the slope of a regression line through the origin. To quantify the net effect of each ablated parameter per observer, we regressed a linear model with zero intercept to $\beta_{\text{fit}}$ versus $\beta_{\text{true}}$. If an ablated parameter has little impact on $\beta_{fit}$, then the slope of the regression will be near 1, so we use 1 minus the linear model's slope as an index of the parameter's contribution. The regression model accounted for errors in both $x$ and $y$ but approximated them as Gaussian. Defining $m$ to be the regression slope for the population and $m_i$ to be the slope for observer $i$, the regression model was defined as

$$\sigma_m \sim \text{half-cauchy}(0,5) \tag{23}$$

$$m_i \sim \mathcal{N}(m, \sigma_m) \tag{24}$$

$$\beta_{\text{true},i} \sim \mathcal{N}(x_i, \sigma_{x,i}) \tag{25}$$

$$\beta_{\text{fit},i} \sim \mathcal{N}(x_i m_i, \sigma_{y,i}) . \tag{26}$$

This model was implemented in STAN and fit using NUTS [86]. The regression was done separately for each experimental condition and each set of ablated parameters. Eqs (23) and (24) are standard practice in hierarchical regression—they capture the idea that there is variation in the parameter of interest (the slope $m$) across observers which is normally distributed with unknown variance, $\sigma_m$, but that this variance is encouraged to be small if supported by the data. The variable $x_i$ is the "true" x location associated with each observer, which is inferred as a latent variable to account for measurement error in both x (Eq (25)) and y (Eq (26)) dimensions. Measurement errors in $\beta_{\text{true}}$, $\sigma_{x,i}$ were set to the standard deviation in $\beta$ across bootstraps. Measurement errors in $\beta_{\text{fit}}$, $\sigma_{y,i}$ were set to the standard deviation of the posterior predictive distribution over $\beta$ from simulated choices on each sample of model parameters as described above. We set $x_i$ and $y_i$ to the median values of $\beta_{true}$ (across bootstrapped trials) and $\beta_{fit}$ (across posterior samples), respectively.

**Ground-truth models.** Based on observations of the temporal weighting profile alone, the transition between primacy and recency could be explained by bounded integration with a changing leak amount in the LSHC condition and high bound in the HSLC condition (Fig G in S1 Text). To verify that all of the above fitting and ablation procedures could distinguish a confirmation bias from bounded integration, we tested them on two ground-truth models:

one where choices were simulated from a hierarchical inference (IS) model, and one where choices were simulated from an ITB model. All ground-truth parameter values are given in Table C in S1 Text, which were chosen to meet two criteria: first, constant performance at 70% in both LSHC and HSLC regimes, and second, matched temporal slopes (a primacy effect with shape $\beta \approx -0.1$ in the LSHC condition and a recency effect with shape $\beta \approx 0.1$ in the HSLC condition for both models). This analysis confirmed that bounded integration is indeed distinguishable from a confirmation bias ($\alpha < 0$), in terms of the quality of the fit (Fig I in S1 Text), different inferred parameter values (Fig J in S1 Text), and the ablation tests (Fig K in S1 Text).

## Supporting information

**S1 Text.** Combined Supplemental Text, Figs A-L, Tables A-C, and Algorithms A-B. **Table A**: **Sensory Information, Category Information, and biases in previous studies**. Justification of placement of example prior studies in Fig 1C and description of stimulus manipulations that will move it to the opposite side of the category–sensory–information space. Each manipulation corresponds to a prediction about how temporal weighting of evidence should change from primacy (red) to flat/recency (blue), or vice versa, as a result. **Table B: Stimulus parameters**. **Table C: Parameters of ground-truth models used to test model-fitting**. SI = sensory information. CI = category information. $\gamma/\alpha$ = leak. S = samples per batch (IS model only). B = bound (ITB model only). $\epsilon$ = integration noise. T = decision temperature. $\lambda$ = lapse rate. **Fig A**: **Temporal kernels for each condition (LSHC and HSLC), and their difference between conditions, for each of four regularization techniques**. In all panels, weights are normalized to have a mean of 1, individual observers are shown as faint thin lines, and the average across observers as a darker bold line. First row ("Logisitic Regression") is the result of ridge regression for predicting choices from per-frame signal levels with no further regularization. Second row ("Smooth Logistic Regression") includes a second-order autoregressive penalty, resulting in smoother kernels. Third row ("Linear Kernels") is a three-parameter model that constrains weights to be a linear function of time. The three parameters control the slope and intercept of the kernel, and the choice bias (Methods). Fourth row ("Exponential Kernels") is a similar three-parameter model that instead constrains weights to be an exponential function of time. **Fig B**: **Cross-validation selects linear or exponential shapes for temporal weights, compared to both unregularized and smoothness-regularized logistic regression**. Panels show 20-fold cross-validation performance of four regression methods to predict choices from sub-threshold trials, separated by task type and by observer. All values are relative to the log-likelihood, per fold, of the unregularized model. Error bars show standard error of the mean difference in performance across folds of shuffled data. "Unregularized LR" refers to standard ridge regression with no regularization of the temporal shape. "Regularized LR" refers to the AR2-penalized logistic regression objective, where the hyperparameters were chosen to maximize cross-validated fitting performance separately for each observer. "Exponential" is the 3-parameter model where weights are an exponential function of time (Eq (3) plus a bias term). Similarly, the "Linear" model constrains the weights to be a linear function of time as in Eq (4), plus a bias term. **Fig C: Comparing exponential and linear regression weights**. Left panel is the same as Fig 5E in the main text, comparing slope of temporal weights by constraining weights to be an exponential function of time. The right panel shows the same analysis with weights constrained to be a linear function of time. In both cases, 9 of 12 observers individually have a significant increase in slope ($p < 0.05$, bootstrap). A one-sided sign test on the medians for each observer reveals a significant population effect with $p = .0032$ (∗∗)for the exponential method and $p = 0.00024$ (∗∗∗) for the linear method. **Fig D: Effect of leak ($\gamma$) parameter in hierarchical inference models**. In both models, larger $\gamma$ increases the prevalence

of recency effects across the entire task space. Panels are as in Fig 3 in the main text. **a-c** sampling model with $\gamma = 0$. **d-f** sampling model with $\gamma = 0.1$. **g-i** sampling model with $\gamma = 0.2$. **j-l** variational model with $\gamma = 0$. **m-o** variational model with $\gamma = 0.1$. **p-r** variational model with $\gamma = 0.2$. **Fig E: Performance of hierarchical inference model using optimal leak ($\gamma$)**. Optimizing performance with respect to $\gamma$ (see also Fig F in S1 Text). **a)** Sampling model performance across task space with $S = 5$ and $\gamma = 0.5$ (compare with Fig 3C in which $\gamma = 0.1$). **b)** Difference in performance for $\gamma = 0.5$ versus $\gamma = 0.1$. Higher $\gamma$ improves performance in the upper part of the space where the confirmation bias is strongest. **c)** Optimizing for performance, the optimal $\gamma^*$ depends on the task. Where the confirmation bias had been strongest, optimal performance is achieved with a stronger leak term. **d)** Model performance when the optimal $\gamma^*$ from (c) is used in each task. **e)** Comparing the ideal observer to (d), the ideal observer still outperforms the model but only in the upper part of the space. **f)** Temporal weight slopes when using the optimal $\gamma^*$ are flat everywhere. The models reproduce the change in slopes seen in the data only when $\gamma$ is fixed across tasks (compare Fig D in S1 Text). **Fig F: Further investigation of optimal leak ($\gamma$)**. Simulation results for optimal leak ($\gamma$) for two further model variations, panels as in Fig E in S1 Text. **a-f** Variational model results. As in the sampling model, we see that the optimal value of $\gamma^*$ increases with category information, or with the strength of the confirmation bias. **h-l** Sampling model results with $S = 1$ (in the main text and Fig E in S1 Text we used $S = 5$). Since the sampling model without a leak term approaches the ideal observer in the limit of $S \rightarrow \infty$, the optimal $\gamma^*$ was close to 0 for much of the space in the main text figure. Here, by comparison, $\gamma^* > 0$ is more common because the $S = 1$ model is more biased. **Fig G: Simulation of bounded integration (ITB) model**. **a)** Performance of an ITB model is not differentially modulated by sensory and category information. **b)** ITB consistently produces primacy effects, as in [7]. **c)** The primacy effect becomes more extreme in regions where evidence is stronger, since the bound is hit earlier in the trial. **d-f)** As in (a-c), but with an additional leak term, resulting in less extreme primacy effects and a transition to recency for *difficult* tasks, but no transition from primacy to recency along the iso-performance contour. (Also note the departure from monotonic exponential-like weight profiles). **g-i)** We now vary the leak term, $\alpha$, as a function of category information. This reproduces the qualitative transition from primacy in LSHC to recency in HSLC. As measured by an exponential fit ($\beta$), slopes are matched to those in the confirmation bias models (Fig 3D and 3G). **Fig H: Simulation results on the larger model of Haefner et al (2016) [42]**. **a)** Performance as a function of sensory information (grating contrast) and category information (probability that each frame matches the trial category). White line is iso-performance contour at 70%, and dots correspond to LSHC and HSLC parameter regimes plotted in (b). Simulation details in S1 Text. **b)** Temporal weights from LSHC and HSLC simulations corresponding to colored points in (a), normalized in each condition so the weights have mean 1. As in the reduced models in the main text, we see a transition from primacy to recency. **Fig I: Results of direct model comparison between IS model and ITB model(s) fit to ground-truth data**. Lower AIC indicates better fit. An ideal integrator (gold) and ground-truth (gray) values serve as upper- and lower-bounds, respectively, on plausible AIC values. In all cases, the best fitting model recovered parameters that are as good as the ground truth. The standard ITB model (with positive leak enforced) is distinguishable from the IS model in the LSHC simulation (top row). However, an Extended ITB model that allows for negative leak (purple), fits all data in all conditions as well as the ground-truth. For this reason, we state in the main text that a negative leak is *functionally* indistinguishable from the true IS model. We pursued *parameter comparison* within this Extended ITB model class, rather than *model comparison* between IS and ITB, in the main text. **Fig J: Box and whisker plots of inferred parameter values**. Showing inferred parameter values in the extended ITB model for each of 12 observers as well as the ground truth models (IS and

ITB—see Table C in S1 Text). Each parameter and observer has two fits, one for the LSHC condition (lower/red) and one for the HSLC condition (upper/blue). Thin lines are 95% posterior interval, thick lines are 50% interval, and points are posterior median. Parameter names are as in the main paper, restated here: $p_C$ = prior over categories, $\lambda$ = symmetric lapse rate, $T$ = decision temperature, $s$ = signal scale (fixed to 1 for ground truth models), $\alpha$ = leak, $B$ = bound, $\epsilon$ = noise. **Fig K**: **Parameter ablation analysis on ground-truth models**. Recall that the ITB model has a primacy effect in the LSHC condition driven by bounded integration. The key signature of bounded integration dynamics is that when the bound is ablated, the leak takes over and it flips to no bias or a recency bias. The key signature of a hierarchical inference model (here, Importance Sampling or IS), on the other hand, is that the primacy bias is unaffected by ablating the bound, but disappears when the leak term is ablated, since a negative leak acts as a confirmation bias. In the HSLC condition (right panel), both models' recency effects are driven by leaky integration. The ITB model's bound competes with the leak, however, so ablating the bound results in exaggerated recency effects, and ablating the leak results in primacy effects. The key signature of a hierarchical inference model, on the other hand, is a recency effect that is unaffected by ablating the bound and that disappears when the leak is ablated. **Fig L**: **Additional information on fits of the Extended ITB model to empirical data and ablation analyses**. **a)** Copy of Fig 6D. Comparing with Fig K in S1 Text suggests that primacy effects are largely driven by confirmation-bias dynamics rather than by bounded integration. **b)** Temporal bias of the full Extended ITB model (x-axis) versus the ablated model (y-axis) for each observer and each ablated parameter in the LSHC condition (each observer has two points at the same x coordinate, offset for visualization). We regressed a single slope for each ablated parameter to summarize the fraction of bias in the population explained by the leak parameter (green) or the bound parameter (purple). **c)** Copy of Fig 6E from the main text. The fact that the leak parameter explains 99.4% of the population primacy effects corresponds to the green regression line being nearly horizontal in (b). **d-f)** Same as (a-c) but for the HSLC condition. As in Fig 6, outlier observer in—who had a *primacy* bias in the HSLC condition—is shown as a diamond symbol in panels (a), (d), and (f). **Algorithm A**: **Hierarchical inference using Importance Sampling**. **Algorithm B**: **Hierarchical inference using Variational Bayes**. (PDF)

## Acknowledgments

We thank Matthew Hochberg for help with initial implementations of the experiment software, and Chris Summerfield for early comments on our experiment design. We also thank Richard Born, Hendrikje Nienborg, Martina Poletti, Duje Tadin, and Ariel Zylberberg for feedback on the manuscript.

## Author Contributions

**Conceptualization:** Richard D. Lange, Ankani Chattoraj, Ralf M. Haefner.

**Data curation:** Richard D. Lange, Ankani Chattoraj.

**Formal analysis:** Richard D. Lange, Ankani Chattoraj, Jeffrey M. Beck, Ralf M. Haefner.

**Funding acquisition:** Ralf M. Haefner.

**Investigation:** Richard D. Lange, Ankani Chattoraj.

**Methodology:** Richard D. Lange, Ankani Chattoraj, Jacob L. Yates, Ralf M. Haefner.

**Resources:** Jacob L. Yates, Ralf M. Haefner.

**Software:** Richard D. Lange, Ankani Chattoraj, Jacob L. Yates.

**Supervision:** Ralf M. Haefner.

**Validation:** Richard D. Lange, Ankani Chattoraj.

**Visualization:** Richard D. Lange, Ankani Chattoraj.

**Writing – original draft:** Richard D. Lange, Ralf M. Haefner.

**Writing – review & editing:** Richard D. Lange, Ankani Chattoraj, Jeffrey M. Beck, Jacob L. Yates, Ralf M. Haefner.

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
