## [Decision Letter · Decision Letter 0]

24 Aug 2021

Dear Dr Haefner,

Thank you very much for submitting your manuscript "A confirmation bias in perceptual decision-making due to hierarchical approximate inference" for consideration at PLOS Computational Biology.

As with all papers reviewed by the journal, your manuscript was reviewed by members of the editorial board and by several independent reviewers. In light of the reviews (below this email), we would like to invite the resubmission of a significantly-revised version that takes into account the reviewers' comments.

Both reviewers found the paper very strong and highly engaging, and provided positive comments. Reviewer 2 had some additional suggestions which I hope you will take into consideration, namely to better clarify the relationship between 'confirmation bias' as it has been variously used in the literature (e.g. temporal weighting vs evidence discounting) and how it is used in the present manuscript, and to place the present manuscript more explicitly within the existing literature in both the introduction and discussion. Reviewer 2 also requests some more details about how specific parameter fitting was performed and what the results of such fitting are, including a request for additional visualizations.

We cannot make any decision about publication until we have seen the revised manuscript and your response to the reviewers' comments. Your revised manuscript is also likely to be sent to reviewers for further evaluation.

Sincerely,

Megan A. K. Peters, Ph.D.

Associate Editor

PLOS Computational Biology

Samuel Gershman

Deputy Editor

PLOS Computational Biology

Reviewer's Responses to Questions

**Comments to the Authors:**

Reviewer #1: This paper tackles a puzzling effect in the literature: why do temporal integration biases (recency vs. primacy) differ across studies of perceptual decision-making? The authors propose an elegant, unified solution where the main driver of such biases is the relative amount of category vs. stimulus noise in each sample of evidence. This follows from a model of hierarchical approximate inference, where the effect of the category prior on sensory likelihood drives primacy effects ('confirmation bias'). Recency effects can be produced when an additional leak term is added to the model. The authors then show that this model can better account for the data than established leaky temporal integration.

This is beautiful work that ties together a great many studies from the literature, and will surely influence how the field thinks about the shape of psychophysical kernels. The paper is well written, the data crystal clear, and the results appealingly presented (I appreciate the indication of which datapoints are authors in Figure 5). I unfortunately lack the mathematical expertise to validate the derivations of the inference model; however, I fully trust the author's expertise in this domain. I was impressed when I first saw this presented at a conference a few years ago, and strongly support publication of this work.

I only have a few small suggestions which may further improve this excellent paper.

- In the intro or discussion, it would be good to add a short discussion of Keung et al. (https://doi.org/10.1038/s41467-020-15630-0) who propose that divisive normalization can account for different kernel shapes. Specifically, it would be interesting to discuss non-linear 'bump' kernels described in this paper, which may be present in some observers in Figure 5a. Could your model account for such unusual shapes?

- l. 145: "... which in turn is determined by the category information in the task". This does not naturally follow - could you add an intuition for how the influence of the prior depends on the stimulus vs. category information in a specific task? Only when reading the legend of Figure 2 repeatedly did the effect of the category information on prior effect become clearer, it may be helpful to briefly state the intuition upfront.

- I find the abbreviations LSHC and HSLC rather confusing, taking a lot of working memory to remember their placement in the space in figure 1c. Please consider replacing them with something easier to parse (perhaps 'sensory noise dominated' vs. 'category noise dominated').

- l. 257 and 267: 6c -> 6b (?), and l. 277: 6c -> 6b

Reviewer #2: The heterogeneity in temporal weighting profiles in laboratory decision-making experiments has been well-documented. Yet little is known about the factors governing this heterogeneity. In this article titled “A confirmation bias in perceptual decision-making due to hierarchical approximate inference”, Lange and colleagues addressed this question suggesting that statistics of the stimuli presented during the task is one such factor. They provided support to this hypothesis with a qualitative clustering of previous studies, and a combination of novel psychophysics task manipulation with human observers and Bayesian modelling.

The article offers a fresh perspective towards the variety of temporal weighting profiles observed in decision-making studies. I believe the perceptual decision-making community finds this insight very relevant, and interesting. I congratulate the authors on a study well-done. I have a few comments most of which can be addressed by revisions to the manuscript without the need for any additional analyses.

Major concerns:

1. The authors use the term “confirmation bias” rather gratuitously throughout the article. It has a widely accepted definition in psychology as the tendency to selectively seek or interpret information to support existing beliefs or hypotheses (Nickerson, 1998). But this is not what the authors imply when they say “perceptual confirmation bias”, but rather a mechanism by which an under-corrected prior lead to primacy. As far as I know, a primacy effect is not selective (where supporting evidence is overweighed or conflicting evidence in underweighted), but a general tendency to ignore evidence once a decision (or a bound, in terms of integration to bound framework) is reached. I would suggest the authors to drop it completely: the article will not meaningfully change as a result of this. If the authors feel strongly about keeping the term, then please clarify how their definition of the term is similar to the generally accepted one. If the authors choose to do this, please spend some space in the discussion relating this work to recent studies investigating confirmation bias in perceptual decisions (Talluri, Urai et al. 2018 Current Biology, and Rollwage et al. 2020 Nat Comm. come to mind).

2. Related to the point above, the title of the article is misleading- I started reading the paper expecting insights into confirmation bias, and found myself wondering how the article adds to our understanding of this bias. I would suggest the authors to use a title more appropriate to what the article is about- which is temporal weighting biases.

3. The article does not mention (or only does so sparingly) some of the recent studies on flexible temporal weighting strategies in perceptual decisions within individuals. Atleast three other studies come to mind:

Levi et al. 2018 eNeuro (doi: 10.1523/ENEURO.0169-18.2018)

Bronfman et al. 2016 PLoS Comp Biol. (doi: 10.1371/journal.pcbi.1004667)

Talluri et al. 2021 J. Neurophys. (doi: 10.1152/jn.00462.2020)

Given the relevance between these studies and the current one, I urge the authors to dedicate a paragraph in the introduction giving an overview of the state of the art in temporal weighting biases in decision-making studies, and how the current study addresses the gaps unfilled by these studies. Furthermore, the similarities and differences between the current article and these studies warrants a paragraph in the discussion. For example, does the approximate hierarchical inference also account for the findings in these previous studies? Can the dynamic LCA model proposed to explain the empirical findings in Bronfman et al. 2016 also account for the empirical findings in the current article?

4. Lines 177-179: “While the exact magnitude of the leak is a free parameter in our model, to be constrained by data, the change in bias with changes in category information is a strong prediction…”. If the leak parameter was indeed fit to the empirical data from the experiment, then please plot this for all subjects as a subfigure in figure 5. This was shown for the extended ITB model in figure 6, but not for the approximate hierarchical inference model.

Minor concerns:

1. This is mostly about semantics, so the authors can choose to ignore it: Primacy/recency biases have been demonstrated in non-perceptual paradigms as well (for eg., Peterson & DuCharme 1967 J Exp. Psych.; Tsetsos et al. 2012 PNAS; Bronfman et al. 2015 Proc R Soc B; Tsetsos et al. 2016 PNAS; Talluri et al. 2021 J Neurophys., to name a few). It helps with the generalizability of the model to these paradigms if the authors replace “sensory information” with another term (like “feature information”, reflecting that features of the stimulus used for the decision-making process) that is not very grounded in the perceptual paradigm.

2. Lines 134-135: “instead, the decision-making area now needs to account for or “divide out” the influence of the top-down prior on the sensory representation (Figure 2b-c).” This sentence sounds like a decision-making area in the brain actively discounts the prior to prevent double counting. Please justify this by citing any relevant past work, or clarify if this is one of the predictions of the current article.

3. Line 358: Please cite Ossmy et al 2013 Current Biology (doi: 10.1016/j.cub.2013.04.039) which showed that humans also adapt their timescale of evidence integration to stimulus statistics.

4. Please include missing citations for Deneve et al. 2012 at line 42, and Prat-Ortega et al. 2021 at line 43.

5. Please update references 57, & 63 to reflect their current publication status. These studies are now published.

**Have the authors made all data and (if applicable) computational code underlying the findings in their manuscript fully available?**

Reviewer #1: **No: **The data are not yet shared publicly, but the authors indicate they will share upon publication. I strongly encourage them to do so.

Reviewer #2: **No: **Neither the empirical data nor the code to reproduce the analyses were made available in the article.

PLOS authors have the option to publish the peer review history of their article (what does this mean?). If published, this will include your full peer review and any attached files.

Reviewer #1: No

Reviewer #2: No
---

## [Editor Report · Decision Letter 1]

1 Oct 2021

Dear Dr Haefner,

We are pleased to inform you that your manuscript 'A confirmation bias in perceptual decision-making due to hierarchical approximate inference' has been provisionally accepted for publication in PLOS Computational Biology.

Best regards,

Megan A. K. Peters, Ph.D.

Associate Editor

PLOS Computational Biology

Samuel Gershman

Deputy Editor

PLOS Computational Biology

Associate Editor's comments:

I have evaluated the authors' revisions and pointy-by-point response to reviews. The authors have adequately addressed all the reviewers' comments and I am now happy to accept this manuscript for publication. I remind the authors that they will need to make their code and data available on the OSF site as linked at the bottom of their page 25 upon publication.

---

## [Editor Report · Acceptance letter]

23 Nov 2021

PCOMPBIOL-D-21-01086R1

A confirmation bias in perceptual decision-making due to hierarchical approximate inference

Dear Dr Haefner,

I am pleased to inform you that your manuscript has been formally accepted for publication in PLOS Computational Biology. Your manuscript is now with our production department and you will be notified of the publication date in due course.

With kind regards,

Livia Horvath
